# Intriguing Properties of Vision Transformers

**Muzammal Naseer**[†⋆]    **Kanchana Ranasinghe**[+⋆]    **Salman Khan**[⋆†]
**Munawar Hayat**[¶]    **Fahad Shahbaz Khan**[⋆§]    **Ming-Hsuan Yang**[‡∘▽]

[†]Australian National University, [⋆]Mohamed bin Zayed University of AI, [+]Stony Brook University,
[¶]Monash University, [§]Linköping University, [‡]University of California, Merced,
[∘]Yonsei University, [▽]Google Research
`muzammal.naseer@anu.edu.au`

## Abstract

Vision transformers (ViT) have demonstrated impressive performance across numerous machine vision tasks. These models are based on multi-head self-attention mechanisms that can flexibly attend to a sequence of image patches to encode contextual cues. An important question is how such flexibility (in attending image-wide context conditioned on a given patch) can facilitate handling nuisances in natural images e.g., severe occlusions, domain shifts, spatial permutations, adversarial and natural perturbations. We systematically study this question via an extensive set of experiments encompassing three ViT families and provide comparisons with a high-performing convolutional neural network (CNN). We show and analyze the following intriguing properties of ViT: (a) Transformers are highly robust to severe occlusions, perturbations and domain shifts, *e.g.,* retain as high as 60% top-1 accuracy on ImageNet even after randomly occluding 80% of the image content. (b) The robustness towards occlusions is not due to texture bias, instead we show that ViTs are significantly less biased towards local textures, compared to CNNs. When properly trained to encode shape-based features, ViTs demonstrate shape recognition capability comparable to that of human visual system, previously unmatched in the literature. (c) Using ViTs to encode shape representation leads to an interesting consequence of accurate semantic segmentation without pixel-level supervision. (d) Off-the-shelf features from a single ViT model can be combined to create a feature ensemble, leading to high accuracy rates across a range of classification datasets in both traditional and few-shot learning paradigms. We show effective features of ViTs are due to flexible and dynamic receptive fields possible via self-attention mechanisms. Code: `https://git.io/Js15X`.

## 1   Introduction

As visual transformers (ViT) attract more interest [1], it becomes highly pertinent to study characteristics of their learned representations. Specifically, from the perspective of safety-critical applications such as autonomous cars, robots and healthcare; the learned representations need to be robust and generalizable. In this paper, we compare the performance of transformers with convolutional neural networks (CNNs) for handling nuisances (*e.g.,* occlusions, distributional shifts, adversarial and natural perturbations) and generalization across different data distributions. Our in-depth analysis is based on *three* transformer families, ViT [2], DeiT [3] and T2T [4] across *fifteen* vision datasets. For brevity, we refer to all the transformer families as ViT, unless otherwise mentioned.

We are intrigued by the fundamental differences in the operation of convolution and self-attention, that have not been extensively explored in the context of robustness and generalization. While convolutions excel at learning local interactions between elements in the input domain (*e.g.,* edges and contour information), self-attention has been shown to effectively learn global interactions (*e.g.,*

35th Conference on Neural Information Processing Systems (NeurIPS 2021).

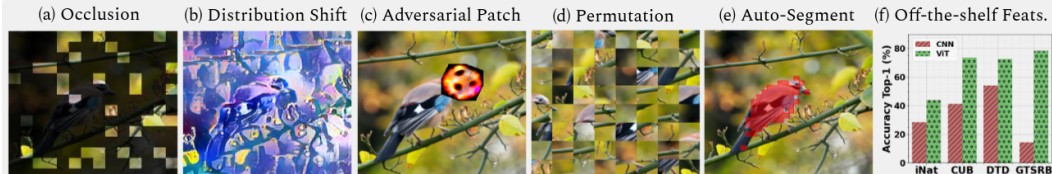

Figure 1: We show intriguing properties of ViT including impressive robustness to (a) severe occlusions, (b) distributional shifts (*e.g.,* stylization to remove texture cues), (c) adversarial perturbations, and (d) patch permutations. Furthermore, our ViT models trained to focus on shape cues can segment foregrounds without any pixel-level supervision (e). Finally, off-the-shelf features from ViT models generalize better than CNNs (f).

relations between distant object parts) [5, 6]. Given a query embedding, self-attention finds its interactions with the other embeddings in the sequence, thereby conditioning on the local content while modeling global relationships [7]. In contrast, convolutions are content-independent as the same filter weights are applied to all inputs regardless of their distinct nature. Given the content-dependent long-range interaction modeling capabilities, our analysis shows that ViTs can flexibly adjust their receptive field to cope with nuisances in data and enhance expressivity of the representations.

Our systematic experiments and novel design choices lead to the following interesting findings:

- ViTs demonstrate strong robustness against severe occlusions for foreground objects, non-salient background regions and random patch locations, when compared with state-of-the-art CNNs. For instance, with a significant random occlusion of up to 80%, DeiT [3] can maintain top-1 accuracy up to ∼60% where CNN has zero accuracy, on ImageNet [8] val. set.
- When presented with texture and shape of the same object, CNN models often make decisions based on texture [9]. In contrast, ViTs perform better than CNNs and comparable to humans on shape recognition. This highlights robustness of ViTs to deal with significant distribution shifts *e.g.,* recognizing object shapes in less textured data such as paintings.
- Compared to CNNs, ViTs show better robustness against other nuisance factors such as spatial patch-level permutations, adversarial perturbations and common natural corruptions (*e.g.,* noise, blur, contrast and pixelation artefacts). However, similar to CNNs [10], a shape-focused training process renders them vulnerable against adversarial attacks and common corruptions.
- Apart from their promising robustness properties, off-the-shelf ViT features from ImageNet pretrained models generalize exceptionally well to new domains *e.g.,* few-shot learning, fine-grained recognition, scene categorization and long-tail classification settings.

In addition to our extensive experimental analysis and new findings, we introduce several novel design choices to highlight the strong potential of ViTs. To this end, we propose an architectural modification to DeiT to encode shape-information via a dedicated token that demonstrates how seemingly contradictory cues can be modeled with distinct tokens within the same architecture, leading to favorable implications such as automated segmentation without pixel-level supervision. Moreover, our off-the-shelf feature transfer approach utilizes an ensemble of representations derived from a single architecture to obtain state-of-the-art generalization with a pre-trained ViT (Fig. 1).

## 2 Related Work

CNNs have shown state-of-the-art performance in independent and identically distributed (i.i.d) settings but remain highly sensitive to distributional shifts; adversarial noise [11, 12], common image corruptions [13], and domain shifts (*e.g.,* RGB to sketches) [14]. It is natural to ask if ViT, that processes inputs based on self-attention, offers any advantages in comparison to CNN. Shao *et al.* [15] analyze ViTs against adversarial noise and show ViTs are more robust to high frequency changes. Similarly, Bhojanapalli *et al.* [16] study ViT against spatial perturbations [15] and its robustness to removal of any single layer. Since ViT processes image patches, we focus on their robustness against patch masking, localized adversarial patches [17] and common natural corruptions. A concurrent work from Paul and Chen [18] also develops similar insights on robustness of ViTs but with a somewhat different set of experiments.

Geirhos *et al.* [9] provide evidence that CNNs mainly exploit texture to make a decision and give less importance to global shape. This is further confirmed by CNN ability to only use local features

[19]. Recently, [20] quantifies mutual information [21] between shape and texture features. Our analysis indicates that large ViT models have less texture bias and give relatively higher emphasis to shape information. ViT's shape-bias approaches human-level performance when directly trained on stylized ImageNet [9]. Our findings are consistent with a concurrent recent work that demonstrates the importance of this trend on human behavioural understanding and bridging the gap between human and machine vision [22]. A recent work [23] shows that self-supervised ViT can automatically segment foreground objects. In comparison, we show how shape-focused learning can impart similar capability in the image-level supervised ViT models, without any pixel-level supervision.

Zeiler *et al*. [24] introduce a method to visualize CNN features at different layers and study the performance of off-the-shelf features. In a similar spirit, we study the generalization of off-the-shelf features of ViT in comparison to CNN. Receptive field is an indication of network's ability to model long range dependencies. The receptive field of Transformer based models covers the entire input space, a property that resembles handcrafted features [25], but ViTs have higher representative capacity. This allows ViT to model global context and preserve the structural information compared to CNN [26]. This work is an effort to demonstrate the effectiveness of flexible receptive field and content-based context modeling in ViTs towards robustness and generalization of the learned features.

# 3 Intriguing Properties of Vision Transformers

## 3.1 Are Vision Transformers Robust to Occlusions?

The receptive field of a ViT spans over the entire image and it models the interaction between the sequence of image patches using self-attention [26, 27]. We study whether ViTs perform robustly in occluded scenarios, where some or most of the image content is missing.

**Occlusion Modeling:** Consider a network $\mathbf{f}$, that processes an input image $x$ to predict a label $y$, where $x$ is represented as a patch sequence with $N$ elements, *i.e.*, $x = \{x_i\}_{i=1}^N$ [2]. While there can be multiple ways to define occlusion, we adopt a simple masking strategy, where we select a subset of the total image patches, $M < N$, and set pixel values of these patches to zero to create an occluded image, $x'$. We refer to this approach as PatchDrop. The objective is then to observe robustness such that $\mathbf{f}(x')_{\text{argmax}} = y$. We experiment with three variants of our occlusion approach, **(a)** Random PatchDrop, **(b)** Salient (foreground) PatchDrop, and **(c)** Non-salient (background) PatchDrop.

*Random PatchDrop:* A subset of $M$ patches is randomly selected and dropped (Fig. 2). Several recent Vision Transformers [2, 3, 4] divide an image into 196 patches belonging to a 14x14 spatial grid; i.e. an image of size 224×224×3 is split into 196 patches, each of size 16×16×3. As an example, dropping 100 such patches from the input is equivalent to losing 51% of the image content.

*Salient (foreground) PatchDrop:* Not all pixels have the same importance for vision tasks. Thus, it is important to study the robustness of ViTs against occlusions of highly salient regions. We leverage a self-supervised ViT model DINO [23] that is shown to effectively segment salient objects. In particular, the spatial positions of information flowing into the final feature vector (class token) within the last attention block are exploited to locate the salient pixels. This allows to control the amount of salient information captured within the selected pixels by thresholding the quantity of attention flow.

We select the subset of patches containing the top $Q\%$ of foreground information (deterministic for fixed $Q$) and drop them. Note that this $Q\%$ does not always correspond to the pixel percentage, *e.g.*, 50% of the foreground information of an image may be contained within only 10% of its pixels.

*Non-salient (background) PatchDrop:* The least salient regions of the image are selected following the same approach as above, using [23]. The patches containing the lowest $Q\%$ of foreground information are selected and dropped here. Note this does not always correspond to the pixel percentage, *e.g.*, 80% of the pixels may only contain 20% of the non-salient information for an image.

Figure 2: An example image with its occluded versions (Random, Salient and Non-Salient). The occluded images are correctly classified by Deit-S [3] but misclassified by ResNet50 [28]. Pixel values in occluded (black) regions are set to zero.
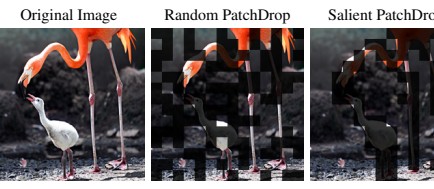
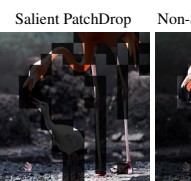

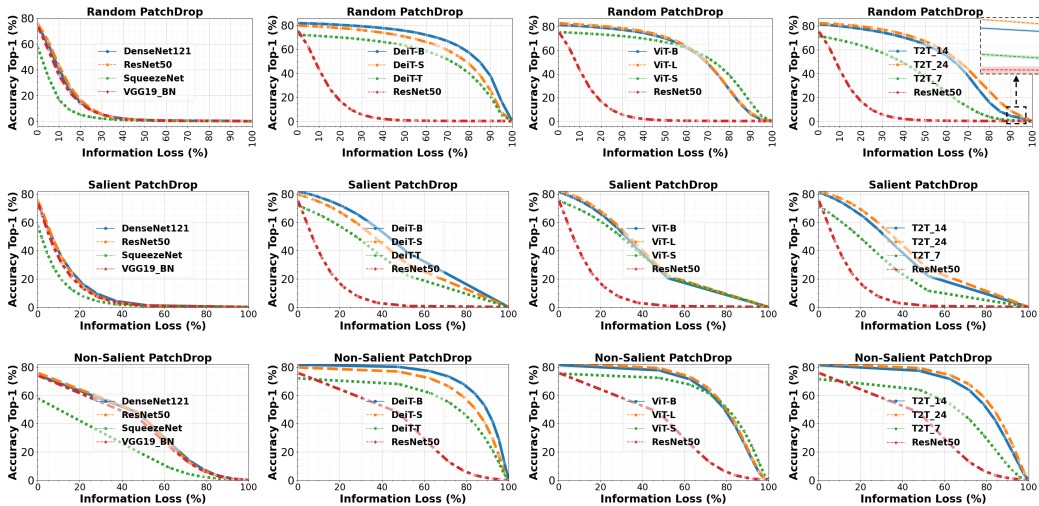

Figure 3: Robustness against object occlusion in images is studied under three PatchDrop settings (see Sec 3.1). (*left*) We study the robustness of CNN models to occlusions, and identify ResNet50 as a strong baseline. (*mid-left*) We compare the DeiT model family against ResNet50 exhibiting their superior robustness to object occlusion. (*mid-right*) Comparison against ViT model family. (*right*) Comparison against T2T model family.

**Robust Performance of Transformers Against Occlusions:** We consider visual recognition task with models pretrained on ImageNet [2]. The effect of occlusion is studied on the validation set (50k images). We define information loss (IL) as the ratio of dropped and total patches ($M$ / $N$). IL is varied to obtain a range of occlusion levels for each PatchDrop methodology. The results (Top-1 %) reported in Fig. 3 show significantly robust performance of ViT models against CNNs. In the case of random PatchDrop, we report the mean of accuracy across 5 runs. For Salient and Non-Salient Patchdrop, we report the accuracy values over a single run, since the occlusion mask is deterministic. CNNs perform poorly when 50% of image information is randomly dropped. For example, ResNet50 (23 Million parameters) achieves 0.1% accuracy in comparison to DeiT-S (22 Million parameters) which obtains 70% accuracy when 50% of the image content is removed. An extreme example can be observed when 90% of the image information is randomly masked but Deit-B still exhibits 37% accuracy. This finding is consistent among different ViT architectures [2, 3, 4]. Similarly, ViTs show significant robustness to the foreground (salient) and background (non-salient) content removal. See Appendix A, B, C, D, and E for further results on robustness analysis.

**ViT Representations are Robust against Information Loss:** In order to better understand model behavior against such occlusions, we visualize the attention (Fig. 4) from each head of different layers. While initial layers attend to all areas, deeper layers tend to focus more on the leftover information in non-occluded regions of an image. We then study if such changes from initial to deeper layers lead to token invariance against occlusion which is important for classification. We measure the correlation coefficient between features/tokens of original and occluded images by using $corr(u,v) = \frac{\sum_i \hat{u}_i \hat{v}_i}{n}$, where $\hat{u}_i = \frac{u_i - E[u_i]}{\sigma(u_i)}$, $E[\cdot]$ and $\sigma(\cdot)$ are mean and standard deviation operations [29]. In our case, random variables $u$ and $v$ refer to the feature maps for an original and occluded image defined over the entire ImageNet validation set. In the case of ResNet50, we consider features before the logit layer and for ViT models, class tokens are extracted from the last transformer block. Class tokens from transformers are significantly more robust and do not suffer much information loss as compared to ResNet50 features (Table 1). Furthermore, we visualize the correlation coefficient across the 12 selected superclasses within ImageNet hierarchy and note that the trend holds across different class types, even for relatively small object types such as *insects*, *food* items and *birds* (Fig. 5). See Appendix F for attention visualizations and G for the qualitative results.

Given the intriguing robustness of transformer models due to dynamic receptive fields and discriminability preserving behaviour of the learned tokens, an ensuing question is whether the learned representations in ViTs are biased towards texture or not. One can expect a biased model focusing only on texture to still perform well when the spatial structure for an object is partially lost.

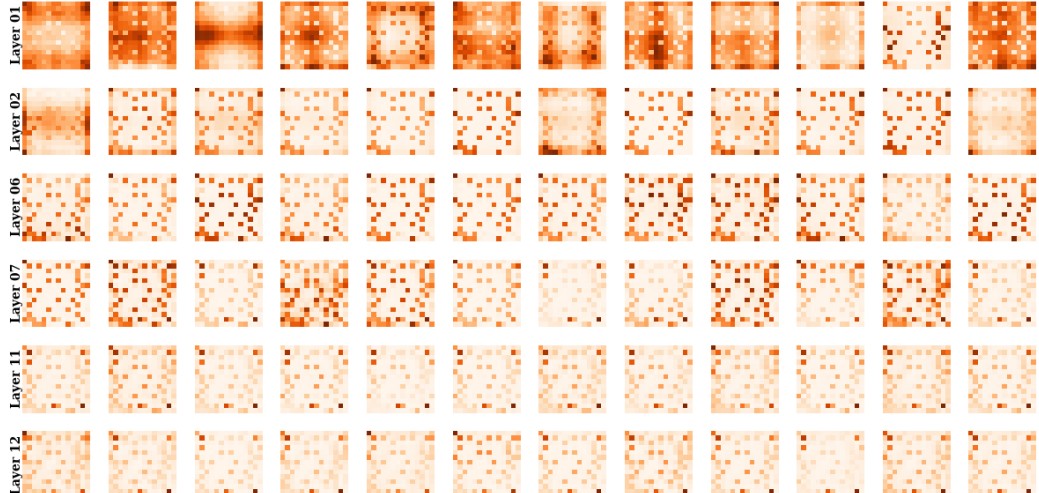

Figure 4: Attention maps (averaged over the entire ImageNet val. set) relevant to each head in multiple layers of an ImageNet pre-trained DeiT-B model. All images are occluded (Random PatchDrop) with the same mask (bottom right). Observe how later layers clearly attend to non-occluded regions of images to make a decision, an evidence of the model's highly dynamic receptive field.

| Model | Correlation Coefficient: Random PatchDrop | | |
|---|---|---|---|
| | 25% Dropped | 50% Dropped | 75% Dropped |
| ResNet50 | 0.32±0.16 | 0.13±0.11 | 0.07±0.09 |
| TnT-S | 0.83±0.08 | 0.67±0.12 | 0.46±0.17 |
| ViT-L | **0.92±0.06** | **0.81±0.13** | 0.50±0.21 |
| Deit-B | 0.90±0.06 | 0.77±0.10 | **0.56±0.15** |
| T2T-24 | 0.80±0.10 | 0.60±0.15 | 0.31±0.17 |

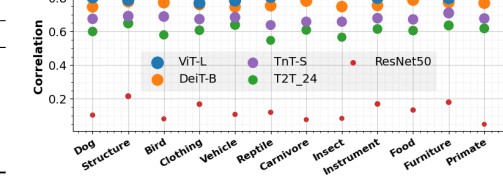

Table 1: Correlation coefficient b/w features/final class tokens of original and occluded images for Random PatchDrop. Averaged across the ImageNet val. set.

Figure 5: Correlation b/w features/final tokens of original and occluded images for 50% Random Drop. Results are averaged across classes for each superclass.

## 3.2 Shape vs. Texture: Can Transformer Model Both Characteristics?

Geirhos *et al*. [9] study shape vs. texture hypothesis and propose a training framework to enhance shape-bias in CNNs. We first carry out similar analysis and show that ViT models preform with a shape-bias much stronger than that of a CNN, and comparably to the ability of human visual system in recognizing shapes. However, this approach results in a significant drop in accuracy on the natural images. To address this issue, we introduce a shape token into the transformer architecture that learns to focus on shapes, thereby modeling both shape and texture related features within the same architecture using a distinct set of tokens. As such, we distill the shape information from a pretrained CNN model with high shape-bias [9]. Our distillation approach makes a balanced trade-off between high classification accuracy and strong shape-bias compared to the original ViT model.

We outline both approaches below. Note that the measure introduced in [9] is used to quantify shape-bias within ViT models and compare against their CNN counterparts.

**Training without Local Texture:** In this approach, we first remove local texture cues from the training data by creating a stylized version of ImageNet [9] named SIN. We then train tiny and small DeiT models [3] on this dataset. Typically, ViTs use heavy data augmentations during training [3]. However, learning with SIN is a difficult task due to less texture details and applying further augmentations on stylized samples distorts shape information and makes the training unstable. Thus, we train models on SIN without applying any augmentation, label smoothing or mixup.

We note that ViTs trained on ImageNet exhibit higher shape-bias in comparison to similar capacity CNN models *e.g.,* DeiT-S (22-Million params) performs better than ResNet50 (23-Million params) (Fig. 6, *right* plot). In contrast, the SIN trained ViTs consistently perform better than CNNs. Interestingly, DeiT-S [3] reaches human-level performance when trained on a SIN (Fig. 6, *left* plot).

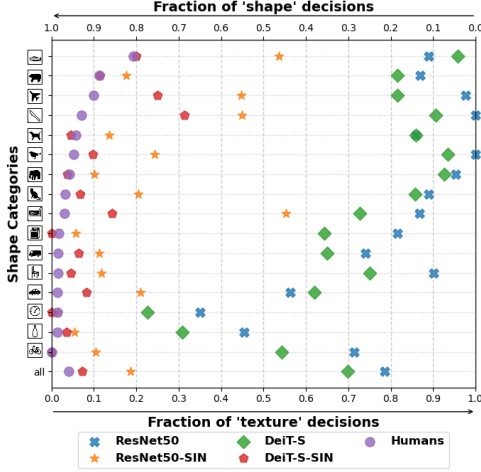

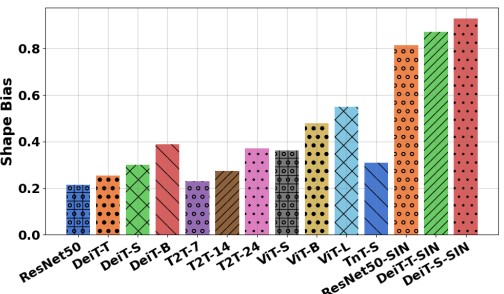

Figure 6: *Shape-bias Analysis:* Shape-bias is defined as the fraction of correct decisions based on object shape. *(Left)* Plot shows shape-texture tradeoff for CNN, ViT and Humans across different object classes. *(Right)* class-mean shape-bias comparison. Overall, ViTs perform better than CNN. The shape bias increases significantly when trained on stylized ImageNet (SIN).

| Model | Distilled | Token Type | ImageNet top-1 (%) | Shape Bias |
|---|---|---|---|---|
| DeiT-T-SIN | ✗ | cls | 40.5 | 0.87 |
| DeiT-T-SIN | ✓ | cls | 71.8 | 0.35 |
| DeiT-T-SIN | ✓ | shape | 63.4 | 0.44 |
| DeiT-S-SIN | ✗ | cls | 52.5 | 0.93 |
| DeiT-S-SIN | ✓ | cls | 75.3 | 0.39 |
| DeiT-S-SIN | ✓ | shape | 67.7 | 0.47 |

Table 2: Performance comparison of models trained on SIN. ViT produces dynamic features that can be controlled by auxiliary tokens. 'cls' represents the class token. During distillation cls and shape tokens converged to vastly different solution using the same features as compared to [3].

**Shape Distillation:** Knowledge distillation allows to compress large teacher models into smaller student models [30] as the teacher provides guidance to the student through soft labels. We introduce a new shape token and adapt attentive distillation [3] to distill shape knowledge from a CNN trained on the SIN dataset (ResNet50-SIN [9]). We observe that ViT features are dynamic in nature and can be controlled by auxiliary tokens to focus on the desired characteristics. This means that a single ViT model can exhibit both high shape and texture bias at the same time with separate tokens (Table 2). We achieve more balanced performance for classification as well as shape-bias measure when the shape token is introduced (Fig. 7). To demonstrate that these distinct tokens (for classification and shape) indeed model unique features, we compute cosine similarity (averaged over ImageNet val. set) between class and shape tokens of our distilled models, DeiT-T-SIN and DeiT-S-SIN, which turns out to be 0.35 and 0.68, respectively. This is significantly lower than the similarity between class and distillation tokens [3]; 0.96 and 0.94 for DeiT-T and Deit-S, respectively. This confirms our hypothesis on modeling distinct features with separate tokens within ViTs, a unique capability that cannot be straightforwardly achieved with CNNs. Further, it offers other benefits as we explain next.

**Shape-biased ViT Offers Automated Object Segmentation:** Interestingly, training without local texture or with shape distillation allows a ViT to concentrate on foreground objects in the scene and ignore the background (Table 3, Fig. 8). This offers an automated semantic segmentation for an image although the model is never shown pixel-wise object labels. That is, shape-bias can be used as self-supervision signals for the ViT model to learn distinct shape-related features that help localize the right foreground object. We note that a ViT trained without emphasis on shape does not perform well (Table 3).

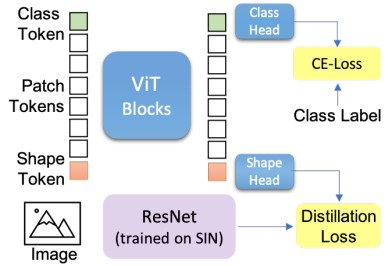

Figure 7: Shape Distillation.

The above results show that properly trained ViT models offer shape-bias nearly as high as the human's ability to recognize shapes. This leads us to question if positional encoding is the key that helps ViTs achieve high performance under severe occlusions (as it can potentially allow later layers to recover the missing information with just a few image patches given their spatial ordering). This possibility is examined next.

| Model | Distilled | Token Type | Jaccard Index |
|---|---|---|---|
| DeiT-T-Random | ✗ | cls | 19.6 |
| DeiT-T | ✗ | cls | 32.2 |
| DeiT-T-SIN | ✗ | cls | 29.4 |
| DeiT-T-SIN | ✓ | cls | 40.0 |
| DeiT-T-SIN | ✓ | shape | 42.2 |
| DeiT-S-Random | ✗ | cls | 22.0 |
| DeiT-S | ✗ | cls | 29.2 |
| DeiT-S-SIN | ✗ | cls | 37.5 |
| DeiT-S-SIN | ✓ | cls | 42.0 |
| DeiT-S-SIN | ✓ | shape | 42.4 |

Table 3: We compute the Jaccard similarity between ground truth and masks generated from the attention maps of ViT models (similar to [23] with threshold 0.9) over the PASCAL-VOC12 validation set. Only class level ImageNet labels are used for training these models. Our results indicate that supervised ViTs can be used for automated segmentation and perform closer to the self-supervised method DINO [23].

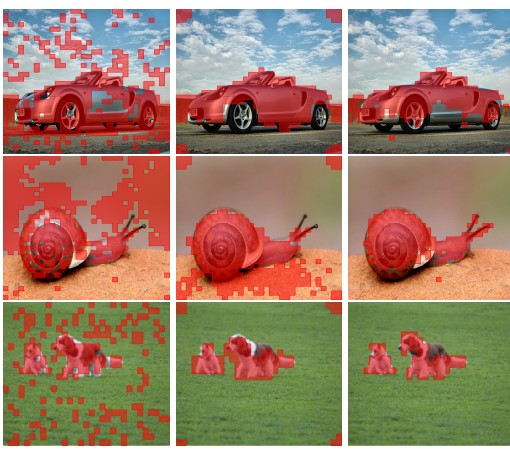

**DeiT-S**   **DeiT-S-SIN**   **DeiT-S-SIN (Distilled)**

Figure 8: Segmentation maps from ViTs. Shape distillation performs better than standard supervised models.

## 3.3 Does Positional Encoding Preserve the Global Image Context?

Transformers' ability to process long-range sequences in parallel using self-attention [27] (instead of a sequential design in RNN [31]) is invariant to sequence ordering. For images, the order of patches represents the overall image structure and global composition. Since ViTs operate on a sequence of images patches, changing the order of sequence *e.g.,* shuffling the patches can destroy the image structure. Current ViTs [2, 3, 4, 26] use positional encoding to preserve this context. Here, we analyze if the sequence order modeled by positional encoding allows ViT to excel under occlusion handling. Our analysis suggests that transformers show high permutation invariance to the patch positions, and the effect of positional encoding towards injecting structural information of images to ViT models is limited (Fig. 10). This observation is consistent with the findings in the language domain [32] as described below.

**Sensitivity to Spatial Structure:** We remove the structural information within images (spatial relationships) as illustrated in Fig. 9 by defining a shuffling operation on input image patches. Fig. 10 shows that the DeiT models [3] retain accuracy better than their CNN counterparts when spatial

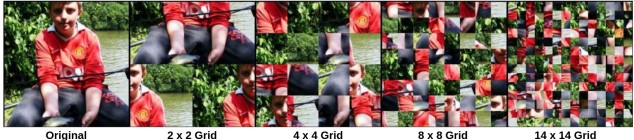

**Original**  **2 x 2 Grid**  **4 x 4 Grid**  **8 x 8 Grid**  **14 x 14 Grid**

Figure 9: An illustration of shuffle operation applied on images used to eliminate their structural information. (*best viewed zoomed-in*)

structure of input images is disturbed. This also indicates that positional encoding is not absolutely crucial for right classification decisions, and the model does not "recover" global image context using the patch sequence information preserved in the positional encodings. Without encoding, the ViT performs reasonably well and achieves better permutation invariance than a ViT using position

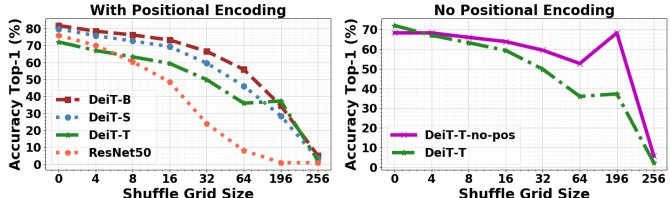

Figure 10: Models trained on 196 image patches. Top-1 (%) accuracy over ImageNet val. set when patches are shuffled. Note the performance peaks when shuffle grid size is equal to the original number of patches used during training, since it equals to only changing the position of input patch (and not disturbing the patch content).

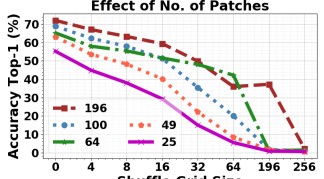

Figure 11: DeiT-T [3] trained on different number of image patches. Reducing patch size decreases the overall performance but also increases sensitivity to shuffle grid size.

| Trained with Augmentations | | | | | | Trained without Augmentation | | | |
|---|---|---|---|---|---|---|---|---|---|
| DeiT-B | DeiT-S | DeiT-T | T2T-24 | TnT-S | Augmix | ResNet50 | ResNet50-SIN | DeiT-T-SIN | DeiT-S-SIN |
| 48.5 | 54.6 | 71.1 | 49.1 | 53.1 | 65.3 | 76.7 | 77.3 | 94.4 | 84.0 |

Table 4: mean Corruption Error (mCE) across common corruptions [13] (lower the better). While ViTs have better robustness compared to CNNs, training to achieve a higher shape-bias makes both CNNs and ViTs more vulnerable to natural distribution shifts. All models trained with augmentations (ViT or CNN) have lower mCE in comparison to models trained without augmentations on ImageNet or SIN.

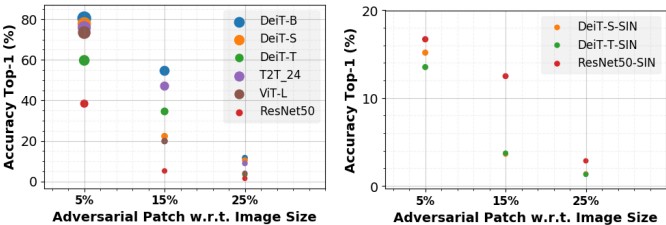

Figure 12: Robustness against adversarial patch attack. ViTs even with less parameters exhibit a higher robustness than CNN. Models trained on ImageNet are more robust than the ones trained on SIN. Results are averaged across five runs of patch attack over ImageNet val. set.

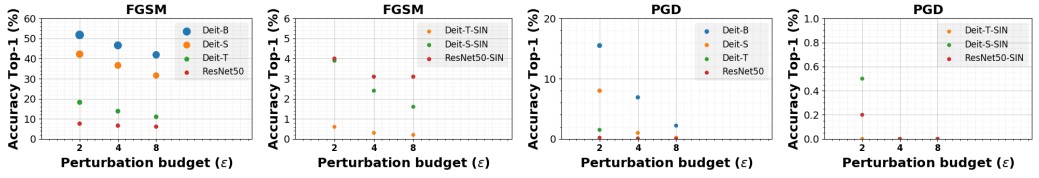

Figure 13: Robustness against sample specific attacks including single step, FGSM [34], and multi-step, PGD [35]. ViTs even with less parameters exhibit a higher robustness than CNN. PGD ran for 5 iterations only. Attacks are evaluated under $l_\infty$ norm and $\epsilon$ represents the perturbation budget by which each pixel is changed in the input image. Results are reported over the ImageNet val. set.

encoding (Fig. 10). Finally, when the patch size is varied during ViT training, the permutation invariance property is also degraded along with the accuracy on unshuffled natural images (Fig. 11). Overall, we attribute the permutation invariance performance of ViTs to their dynamic receptive field that depends on the input patch and can adjust attention with the other sequence elements such that moderately shuffling the elements does not degrade the performance significantly.

The above analysis shows that just like the texture-bias hypothesis does not apply to ViTs, the dependence on positional encodings to perform well under occlusions is also incorrect. This leads us to the conclude that ViTs robustness is due to its flexible and dynamic receptive field (see Fig. 4) which depends on the content of an input image. We now delve further deep into the robustness of ViT, and study its performance under adversarial perturbations and common corruptions.

### 3.4  Robustness of Vision Transformers to Adversarial and Natural Perturbations

After analyzing the ability of ViTs to encode shape information (Sec. 3.2), one ensuing question is: *Does higher shape-bias help achieve better robustness?* In Table 4, we investigate this by calculating mean corruption error (mCE) [13] on a variety of synthetic common corruptions (*e.g.,* rain, fog, snow and noise). A ViT with similar parameters as CNN (*e.g.,* DeiT-S) is more robust to image corruptions than ResNet50 trained with augmentations (Augmix [33]). Interestingly, CNNs and ViTs trained without augmentations on ImageNet or SIN are more vulnerable to corruptions. These findings are consistent with [10], and suggest that augmentations improve robustness against common corruptions.

We observe similar performance against untargeted, universal adversarial patch attack [17] and sample specific attacks including single step, fast gradient sign method (FGSM) [34], and multi-step projected gradient attack known as PGD [35]. Adversarial patch attack [17] is unbounded that is it can change pixel values at certain location in the input image by any amount, while sample specific attacks [34, 35] are bounded by $l_\infty$ norm with a perturbation budget $\epsilon$, where $\epsilon$ represents the amount by which each pixel is changed in the entire image. ViTs and CNN trained on SIN are significantly

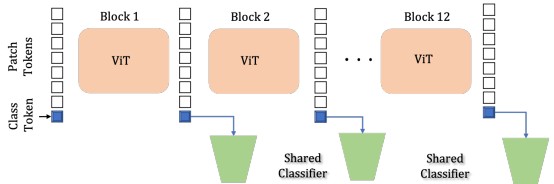

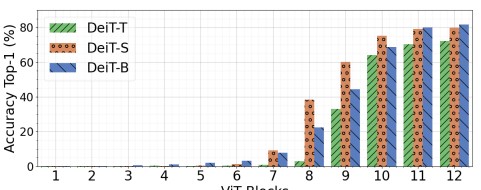

Figure 14: A single ViT model can provide a features ensemble since class token from each block can be processed by the classifier independently. This allows us to identify the most discriminative tokens useful for transfer learning.

Figure 15: Top-1 (%) for ImageNet val. set for class tokens produced by each ViT block. Class tokens from the last few layers exhibit highest performance indicating the most discriminative tokens.

| Blocks | Class Tokens | Patch Tokens | CUB [37] | Flowers [38] | iNaturalist [39] |
|---|---|---|---|---|---|
| Only 12th (last block) | ✓ | ✗ | 68.16 | 82.58 | 38.28 |
| | ✓ | ✓ | 70.66 | 86.58 | 41.22 |
| From 1st to 12th | ✓ | ✗ | 72.90 | **91.38** | 44.03 |
| | ✓ | ✓ | 73.16 | 91.27 | 43.33 |
| From 9th to 12th | ✓ | ✗ | **73.58** | 90.00 | **45.15** |
| | ✓ | ✓ | 73.37 | 90.33 | 45.12 |

Table 5: Ablative Study for off-the-shelf feature transfer on three datasets using ImageNet pretrained DeiT-S [3]. A linear classifier is learned on only a concatenation of class tokens or the combination of class and averaged patch tokens at various blocks. We note class token from blocks 9-12 are most discriminative (Fig. 15) and have the highest transferability in terms of Top-1 (%) accuracy.

more vulnerable to adversarial attack than models trained on ImageNet (Figs. 12 and 13), due to the shape-bias vs. robustness trade-off [10].

Given the strong robustness properties of ViT as well as their representation capability in terms of shape-bias, automated segmentation and flexible receptive field, we analyze their utility as an off-the-shelf feature extractor to replace CNNs as the default feature extraction mechanism [36].

## 3.5 Effective Off-the-shelf Tokens for Vision Transformer

A unique characteristic of ViT models is that each block within the model generates a class token which can be processed by the classification head separately (Fig. 14). This allows us to measure the discriminative ability of each individual block of an ImageNet pre-trained ViT as shown in Fig. 15. Class tokens generated by the deeper blocks are more discriminative and we use this insight to identify an effective ensemble of blocks whose tokens have the best downstream transferability.

**Transfer Methodology:** As illustrated in Fig. 15, we analyze the block-wise classification accuracy of DeiT models and determine the discriminative information is captured within the class tokens of the last few blocks. As such, we conduct an ablation study for off-the-shelf transfer learning on fine-grained classification dataset CUB [37], Flowers [38] and large scale iNaturalist [39] using DeiT-S [3] as reported in Table 5. Here, we concatenate the class tokens (optionally combined with average patch tokens) from different blocks and train a linear classifier to transfer the features to downstream tasks. Note that a patch token is generated by averaging along the patch dimension. The scheme that concatenate class tokens from the last four blocks shows the best transfer learning performance. We refer to this transfer methodology as DeiT-S (ensemble). Concatenation of both class and averaged patch tokens from all blocks helps achieve similar performance compared to the tokens from the last four blocks but requires significantly large parameters to train. We find some exception to this on the Flower dataset [38] where using class tokens from all blocks have relatively better improvement (only 1.2%), compared to the class tokens from the last four blocks (Table 5). However, concatenating tokens from all blocks also increases the number of parameters e.g., transfer to Flowers from all tokens has 3 times more learnable parameters than using only the last four tokens. We conduct further experimentation with DeiT-S (ensemble) across a broader range of tasks to validate our hypothesis. We further compare against a pre-trained ResNet50 baseline, by using features before the logit layer.

`Visual Classification:` We analyze the transferability of off-the-shelf features across several datasets including Aircraft [40], CUB [37], DTD [41], GTSRB [42], Fungi [43], Places365 [44] and iNaturalist [39]. These datasets are developed for fine-grained recognition, texture classification,

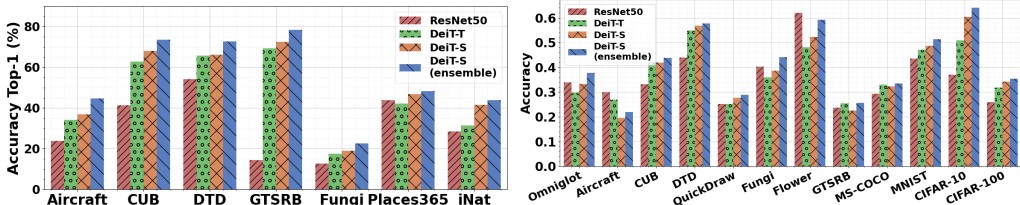

Figure 16: Off-the-shelf ViT features transfer better than CNNs. We explore transferability of learned representations using generic classification as well as few-shot classification for out-of-domain tasks. In the case of classification (*left*), the ImageNet pre-trained ViTs transfer better than their CNN counterparts across tasks. In the case of few-shot learning (*right*), ImageNet pre-trained ViTs perform better on average.

traffic sign recognition, species classification and scene recognition with 100, 200, 47, 43, 1394, 365 and 1010 classes respectively. We train a linear classifier on top of the extracted features over the train split of each dataset, and evaluate the performance on their respective test splits. The ViT features show clear improvements over the CNN baseline (Fig. 16). We note that DeiT-T, which requires about 5 times fewer parameters than ResNet50, performs better among all datasets. Furthermore, the model with the proposed ensemble strategy achieves the best results across all datasets.

`Few-Shot Learning`: We consider meta-dataset [45] designed as a large-scale few-shot learning (FSL) benchmark containing a diverse set of datasets from multiple domains. This includes letters of alphabets, hand-drawn sketches, images of textures, and fine-grained classes making it a challenging dataset involving a domain adaption requirement as well. We follow the standard setting of training on ImageNet and testing on all other datasets which are considered as the downstream tasks.

In our experiments, we use a network pre-trained for classification on ImageNet dataset to extract features. For each downstream dataset, under the FSL setting, a support set of labelled images is available for every test query. We use the extracted features to learn a linear classifier over the support set for each query (similar to [46]), and evaluate using the standard FSL protocol defined in [45]. This evaluation involves a varying number of shots specific for each downstream dataset. On average, the ViT features transfer better across these diverse domains (Fig. 16) in comparison to the CNN baseline. Furthermore, we note that the transfer performance of ViT is further boosted using the proposed ensemble strategy. We also highlight the improvement in QuickDraw, a dataset containing hand-drawn sketches, which aligns with our findings on improved shape-bias of ViT models in contrast to CNN models (see Sec. 3.2 for elaborate discussion).

## 4 Discussion and Conclusions

In this paper, we analyze intriguing properties of ViTs in terms of robustness and generalizability. We test with a variety of ViT models on fifteen vision datasets. All the models are trained on 4 V100 GPUs. We demonstrate favorable merits of ViTs over CNNs for occlusion handling, robustness to distributional shifts and patch permutations, automatic segmentation without pixel supervision, and robustness against adversarial patches, sample specific adversarial attacks and common corruptions. Moreover, we demonstrate strong transferability of off-the-shelf ViT features to a number of downstream tasks with the proposed feature ensemble from a single ViT model. An interesting future research direction is to explore how the diverse range of cues modeled within a single ViT using separate tokens can be effectively combined to complement each other. Similarly, we found that ViTs auto-segmentation property stems from their ability to encode shape information. We believe that integrating our approach and DINO [23] is worth exploring in the future. To highlight few open research questions: a) Can self-supervision on stylized ImageNet (SIN) improve segmentation ability of DINO?, and b) Can a modified DINO training scheme with texture (IN) based local views and shape (SIN) based global views enhance (and generalize) its auto-segmentation capability?

Our current set of experiments are based on ImageNet (ILSVRC'12) pre-trained ViTs, which pose the risk of reflecting potential biases in the learned representations. The data is mostly Western, and encodes several gender/ethnicity stereotypes with under-representation of certain groups [47]. This version of the ImageNet also poses privacy risks, due to the unblurred human faces. In future, we will use a recent ImageNet version which addresses the above issues [48].

## Acknowledgments

This work is supported in part by NSF CAREER grant 1149783, and VR starting grant (2016-05543). M. Hayat is supported by Australian Research Council DECRA fellowship DE200101100.

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
