# A    Random PatchDrop: Effect of Patch Size

We extend our Random PatchDrop experiments to include varying patch sizes for the masking operation, as illustrated in Fig. 17. The PatchDrop experiments in the main paper involved splitting the image into a 14×14 grid (obtaining 196 patches of dimension 16×16 pixels). Here, we split the image into different grid sizes and we define each experiment by the relevant grid size. Results for these experiments are presented in Fig. 18. All accuracy values are reported on the ImageNet val set. Since each grid size contains a different number of patches, we occlude a particular percentage and interpolate to the same scale in our accuracy plots for better comparison.

We note that ViT models (that split an input image into a sequence of patches for processing) are significantly more robust to patch occlusion when dimensions of occluded patches are multiples of the model patch size (the grid size used is a factor of the original grid size). This is visible in the higher performance of ViT for the 7×7 grid PatchDrop experiment (original uses 14×14 grid). At the same time, as large portions are occluded (*e.g.*, with a 4×4 spatial grid), the performance difference between ViT models and CNNs reduces considerably. We believe this to be the case since very large patch occlusions at high masking rates is likely to remove all visual cues relevant to a particular object category, make it really challenging for both ViT and CNN models to make correct predictions.

More importantly, we note that the trends observed in Sec. 3.1 about occlusions are reconfirmed from the varying grid-sizes experiment as well. We also note that some of these grid sizes (*e.g.*, 8×8) have no relation to the grid patterns used by the original ViT models (that split an image into a sequence of 14×14 patches). This indicates that while these trends are more prominent for matching grid sizes (same as that of ViT models) and its factors, the observed trends are not arising solely due to the ViT models' grid operation. We note this behaviour is possible due to the dynamic receptive field of ViTs.

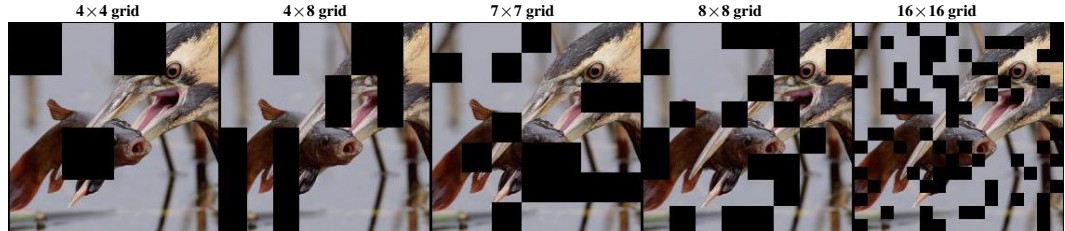

Figure 17: Visualization of varying grid sizes (resulting in different patch sizes) for PatchDrop experiments.

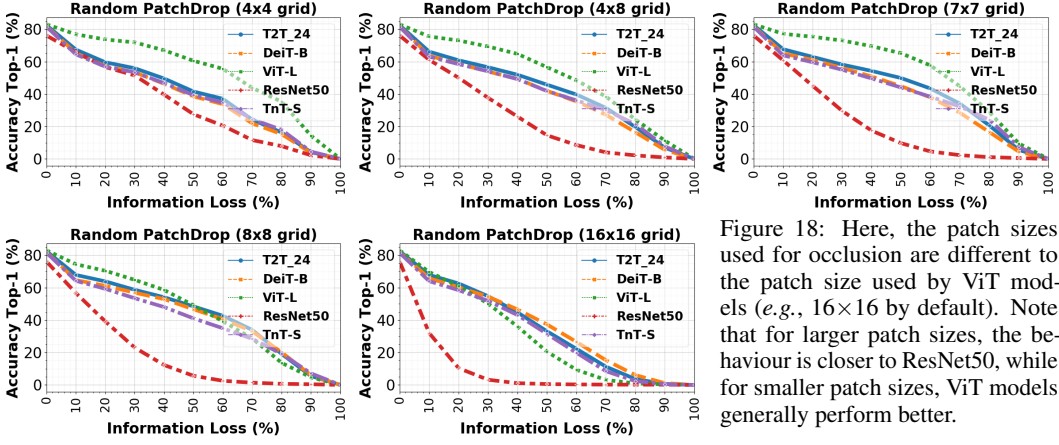

Figure 18: Here, the patch sizes used for occlusion are different to the patch size used by ViT models (*e.g.*, 16×16 by default). Note that for larger patch sizes, the behaviour is closer to ResNet50, while for smaller patch sizes, ViT models generally perform better.

## A.1    Random PatchDrop with Offset

We also explore how a spatial offset on our PatchDrop masks affects ViT models. This is aimed at eliminating the possible alignments between the intrinsic grid patterns of ViT models and our occlusion strategy, thereby avoiding any biasness in the evaluation setting towards a particular model family. The same masks are applied on the image, except with a small spatial offset to ensure that

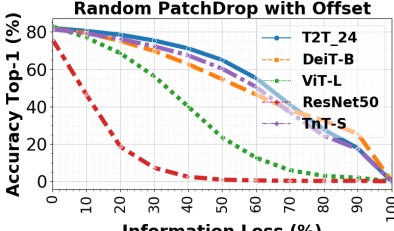

Figure 19: We repeat our experiments in Sec. 3.1 by adding an offset to the grid we use for masking patches. We aim to eliminate any biases due to any gird patterns that bear similarity with the kind of patches used by ViT models. To this end, in the PatchDrop experiments we remove alignment between our masks and ViT grid patterns. We note similar trends in this case as well, alongside a relative drop in ViT-L performance.

none of the masking patches align with any of the grid patterns used by ViT models in processing the input image. We replicate the same experiments as described in Sec. 3.1 under this setting and present our results in Fig. 19. While in general we observe a similar trend between ViT models and the ResNet50 model, we note the significant drop of accuracy in ViT-L, in comparison to its performance under the no-offset setting. We present our potential reasons for this trend below.

ViT-L is a large-scale model containing over 300 million trainable parameters, while the other models contain significantly less parameters *e.g.*, DeiT-B (86 million), T2T-24 (64 million), TnT-S (23 million), and ResNet50 (25 million). Furthermore, unlike ViT-L model, DeiT family and those building on it are trained with extensive data augmentation methods that ensure stable training of ViTs with small datasets. A similar relative drop of ViT-L performance is observed in the 16×16 grid size experiment in Fig. 18 as well. The anomalous behaviour of ViT-L in this setting is potentially owing to these differences.

# B   Random PixelDrop

A further step to observe the occlusion effect decoupled from the intrinsic grid operation of ViT models is to occlude at a pixel level. We generate pixel level masks of varying occlusion levels as illustrated in Fig. 20. Our evaluations on the ImageNet val. set presented in Fig. 21 indicate the same trends between ViT models and CNNs that are observed earlier in Sec. 3.1 and Appendix A.

PixelDrop can be considered as a version of PatchDrop where we use a grid size equal to the image dimensions (setting patch size to 1×1). Considering this, we compare how the performance of models varies as we approach PixelDrop from smaller grid sizes. This is illustrated in Fig. 22 where we evaluate models on the ImageNet val set at 50% occlusion using PatchDrop with different grid sizes. We note that the overall performance of models drops for such fixed occlusion levels in the case of PixelDrop in comparison to the PatchDrop experiments.

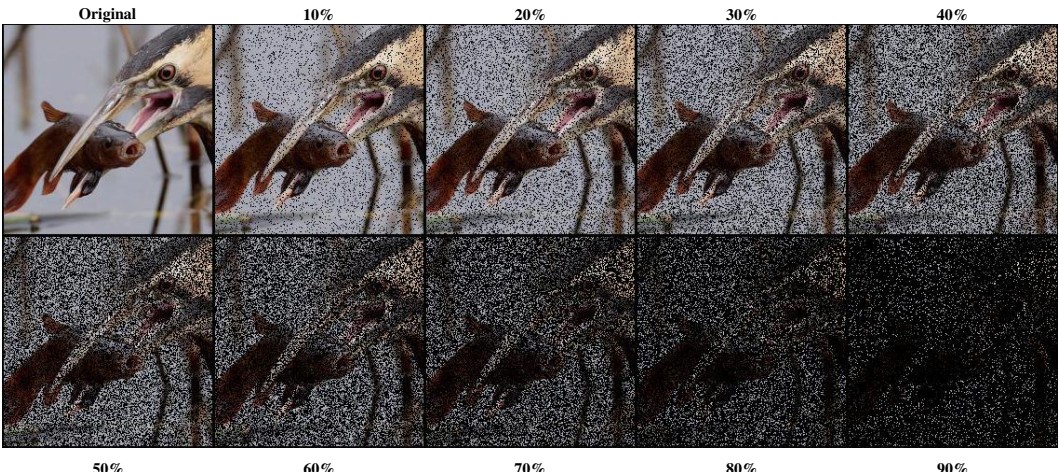

Figure 20: Visualization of varying levels of PixelDrop (randomly masking pixels to study robustness against occlusions).

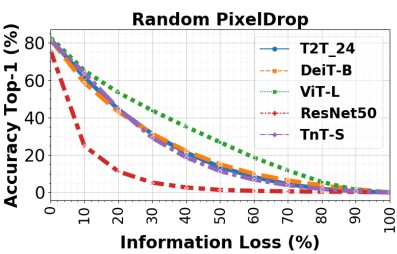

Figure 21: Random PixelDrop: we compare the performance of ViT models against a ResNet50 for our PixelDrop experiments illustrating how similar trends are exhibited.

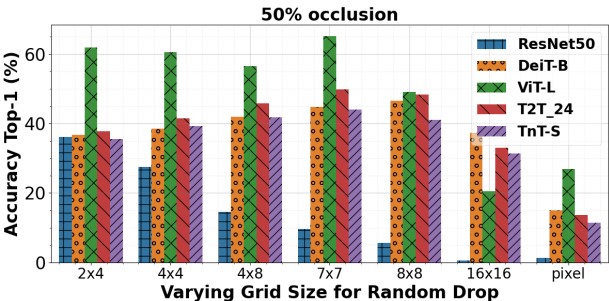

Figure 22: We compare the performance of models as we vary the grid size keeping the occlusion level constant at 50% all the way until PixelDrop which we consider as PatchDrop with grid size equivalent to the image dimensions. While PixelDrop shows us similar trends as the occlusion level varies (Fig, 21), the general performance of models decreases.

We also note how ViT-L displays significantly higher performance in comparison to the other models. This can be attributed to its much higher trainable parameter count as discussed in Sec. A.1. At the same time, ViT-L shows an anomalous drop in performance for the 16×16 grid, quite similar to our observations in Fig. 19.

## C  Robustness to Feature Drop

In contrast to our previous experiments involving occlusion in the model input space, we now focus on occlusion within the model feature space. We achieve this by dropping portions of the intermediate representations inside the ViT model as opposed to dropping patches from the input image. For each transformer block (*e.g.* for each of the 12 blocks in DeiT-B), we randomly mask (set to zero) a selected percentage of its input features. The effects of these "feature drop" experiments are studied in Table 6 by evaluating performance on the ImageNet val set. Performance is measured in the standard method (using the output of the final classifier head of the ViT model). We note that for small amounts of feature drop (25% and 50%), the models suffer relatively similar performance drops regardless of the individual block location. However, for larger amounts of feature drop, certain blocks emerge more important for each model. Furthermore, we notice a level of information redundancy within the blocks of larger models, as their performance drops are not significant even for considerable amounts of feature drop (e.g. ViT-L at 25%).

| Block | ViT-L | | | DeiT-B | | |
|---|---|---|---|---|---|---|
| | 25% | 50% | 75% | 25% | 50% | 75% |
| Block 1 | 75.72 | 67.62 | 25.99 | 57.36 | 38.51 | 15.17 |
| Block 3 | 74.74 | 66.86 | 28.89 | 48.46 | 32.61 | 11.60 |
| Block 5 | 73.32 | 60.56 | 29.69 | 54.67 | 40.70 | 14.10 |
| Block 7 | 75.56 | 69.65 | 53.42 | 55.44 | 43.90 | 24.10 |
| Block 9 | 76.16 | 70.59 | 42.63 | 50.54 | 28.43 | 18.21 |
| Block 11 | 76.28 | 66.15 | 28.95 | 61.97 | 35.10 | 10.94 |

Table 6: Lesion Study: we drop a percentage of features input to each block of selected ViT models and evaluate their performance in terms of Top-1 accuracy (%) on ImageNet val set. ViT-L shows significant robustness against such feature drop even up to the 25% mark hinting towards information redundancy within the model.

| Block | 25% | 50% | 75% |
|---|---|---|---|
| 1 | 0.14 | 0.09 | 0.05 |
| 2 | 45.09 | 4.91 | 0.23 |
| 3 | 69.19 | 28.35 | 0.52 |
| 4 | 73.95 | 64.12 | 18.95 |
| 5 | 75.74 | 75.21 | 73.57 |

Table 7: ResNet50 Lesion Study: we perform feature drop on the intermediate feature maps input to each of the four residual blocks (layers 1-4) and the feature map prior to the final average pooling operation (layer 5). We evaluate Top-1 accuracy (%) on the ImageNet val. set for 25%, 50%, and 75% feature drop applied to each layer.

In Table 7, we conduct the same feature drop experiments for a ResNet50 model. We note that the ResNet50 architecture is entirely different to that of ViT models; hence comparison of these values will give little meaning. In the case of ResNet50, we observe how feature drop in the earlier layers leads to significant drops in performance unlike in ViT models. Also, feature drop in the last layer shows almost negligible drops in performance, which may be due to the average pooling operation which immediately processes those features. In the case of the ViT models compared, the patch tokens in the last layer are not used for a final prediction, so applying feature drop on them has no effect on the performance.

## D    Robustness to Occlusions: More Analysis

In our experimental settings, we used ViTs with class tokens that interact with patch tokens throughout the network and are subsequently used for classification. However, not all ViT designs use a class token e.g., Swin Transformer [49] uses an average of all tokens. To this end, we conduct experiments (Fig. 23) using three variants of the recent Swin Transformer [49] against our proposed occlusions.

### D.1    Swin Transformer [49]

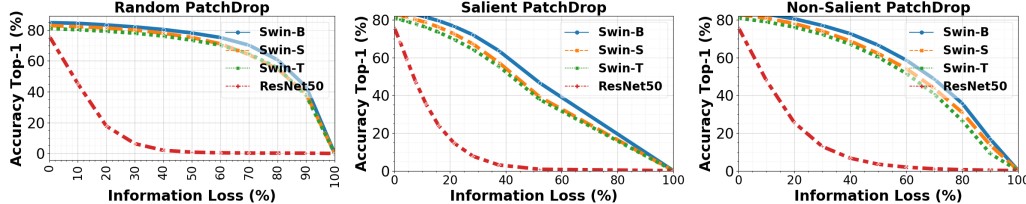

Figure 23: Robustness against object occlusion in images is studied under three PatchDrop settings (see Sec 3.1). We compare the Swin model family against ResNet50 exhibiting their superior robustness to object occlusion. These results show that ViT architectures that does not depend on using explicit class token like Swin transformer [49] are robust against information loss as well.

### D.2    RegNetY [50]

Here, we evaluate three variants of RegNetY against our proposed occlusions (Fig. 24). RegNetY [50] shows relatively higher robustness when compared to ResNet50, but overall behaves similar to other CNN models.

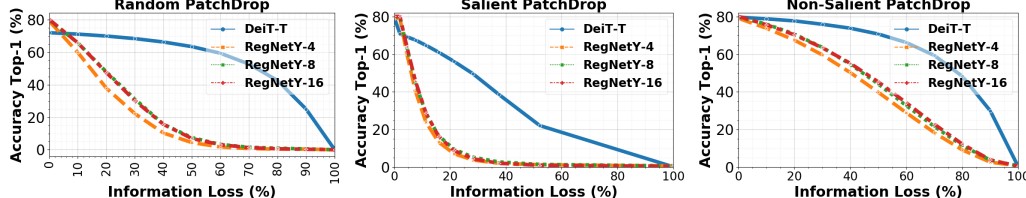

Figure 24: Robustness against object occlusion in images is studied under three PatchDrop settings (see Sec 3.1). We study the robustness of stronger baseline CNN model, RegNetY [50] to occlusions, and identify that it overall behaves similar to other CNN models. Deit-T [3], a ViT with small number of parameters, performs significantly better than all the considered RegNetY variants.

# E  Behaviour of Shape Biased Models

In this section, we study the effect of our PatchDrop (Sec. 3.1) and permutation invariance (Sec. 3.3) experiments on our models trained on Stylized ImageNet [9] (shape biased models). In comparison to a shape biased CNN model, the VIT models showcase favorable robustness to occlusions presented in the form of PatchDrop. Note that ResNet50 (25 million) and DeiT-S (22 million) have similar trainable parameter counts, and therein are a better comparison. Furthermore, we note that in the case of "random shuffle" experiments, the ViT models display similar (or lower) permutation invariance in comparison to the CNN model. These results on random shuffle indicate that the lack of permutation invariance we identified within ViT models in Sec. 3.3 is somewhat overcome in our shape biased models.

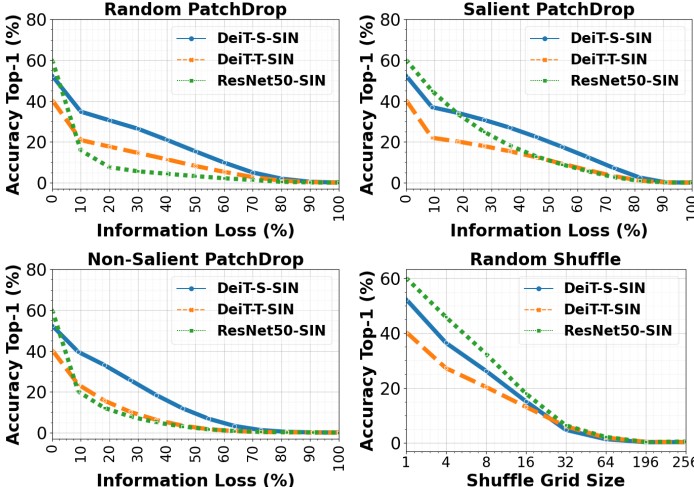

Figure 25: Shape biased models: We conduct the same Patch-Drop and Random Shuffle experiments on DeiT models trained on Stylized ImageNet [9] and compare with a CNN trained on the same dataset. All results are calculated over the ImageNet val. set. We highlight the improved performance in the PatchDrop experiments for the DeiT models in comparsion to ResNet50. We also note how the DeiT models' performance drop with random shuffling is similar to that of the ResNet model.

# F  Dynamic Receptive field

We further study the ViT behavior to focus on the informative signal regardless of its position. In our new experiment, during inference, we rescale the input image to 128x128 and place it within black background of size 224x224. In other words, rather than removing or shuffling image patches, we reflect all the image information into few patches. We then move the position of these patches to the upper/lower right and left corners of the background. On average, Deit-S shows 62.9% top-1 classification accuracy and low variance (62.9±0.05). In contrast, ResNet50 achieves only 5.4% top-1 average accuracy. These results suggest that ViTs can exploit discriminative information regardless of its position (Table 8). Figure 26 shows visualization depicting the change in attention, as the image is moved within the background.

| Models | top-right | top-left | bottom-right | bottom-left |
|--------|-----------|----------|--------------|-------------|
| ResNet50 | 5.59 | 5.71 | 4.86 | 5.30 |
| DeiT-T | 51.21 | 51.38 | 50.61 | 50.70 |
| DeiT-S | 63.14 | 63.01 | 62.62 | 62.79 |
| DeiT-B | **69.37** | **69.29** | **69.18** | **69.20** |

Table 8: We rescale the input image to 128x128 and place it within the upper/lower right and left corners of the background of size 224x224. ViTs can exploit discriminative information regardless of its position as compared to ResNet50. Top-1 (%) accuracy on ImageNet val. set is reported.

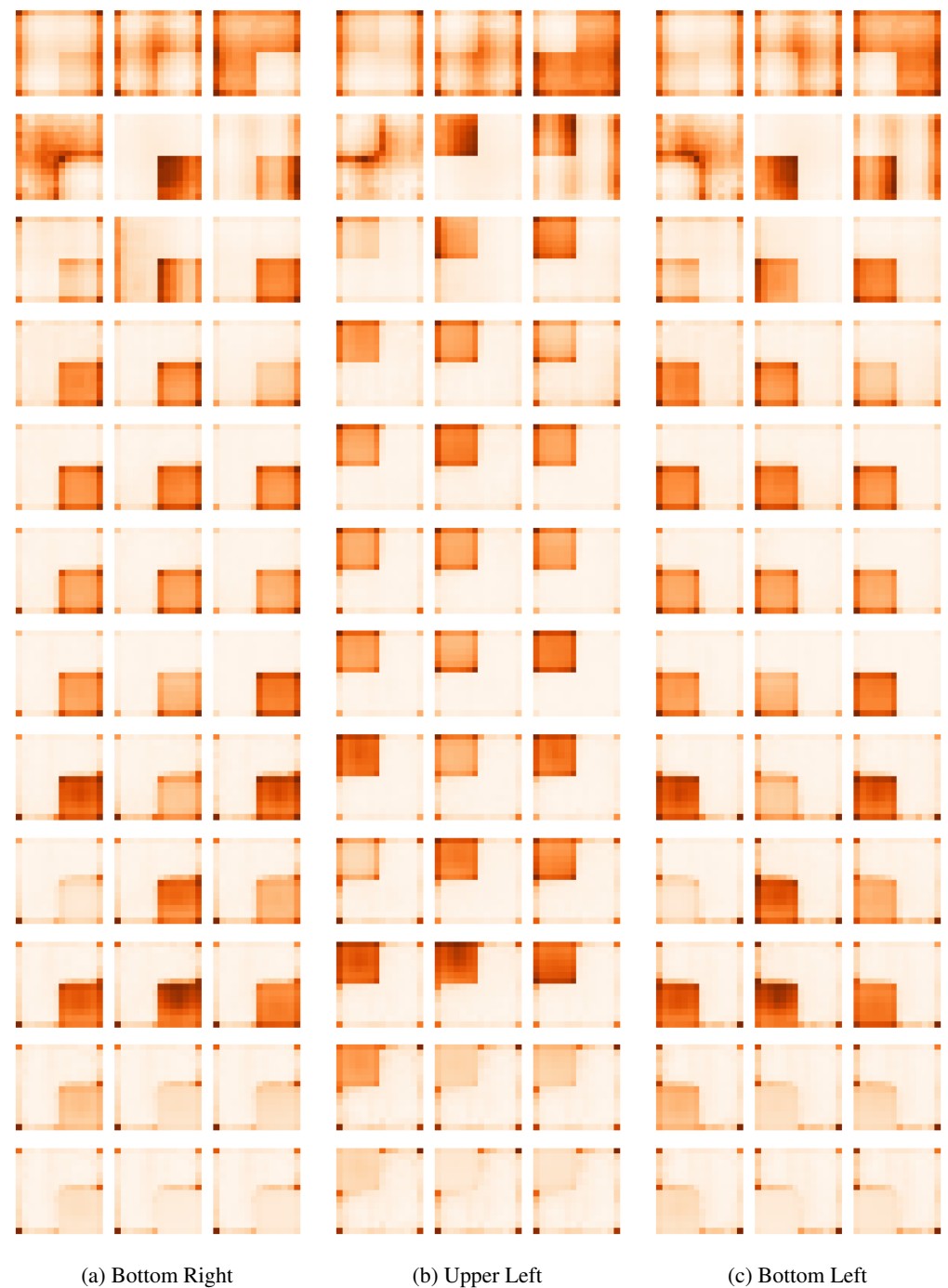

(a) Bottom Right          (b) Upper Left          (c) Bottom Left

Figure 26: Visualization depicting the change in attention, as the image is moved within the background. Attention maps (averaged over the entire ImageNet val. set) relevant to each head across all 12 layers of an ImageNet pre-trained DeiT-T (tiny) model [3]. All images are rescaled to 128x128 and placed within black background. Observe how later layers clearly attend to non-occluded regions of images to make a decision, an evidence of the model's highly dynamic receptive field.

## G    Additional Qualitative Results

Here, we show some qualitative results, *e.g.,* Fig. 27 show the examples of our occlusion (random, foreground, and background) method. The performance of our shape models to segment the salient

image is shown in Fig. 28. We show different variation levels of Salient PatchDrop on different images in Fig. 29. Finally, we show adversarial patches optimized to fool different ViT models (Fig. 31).

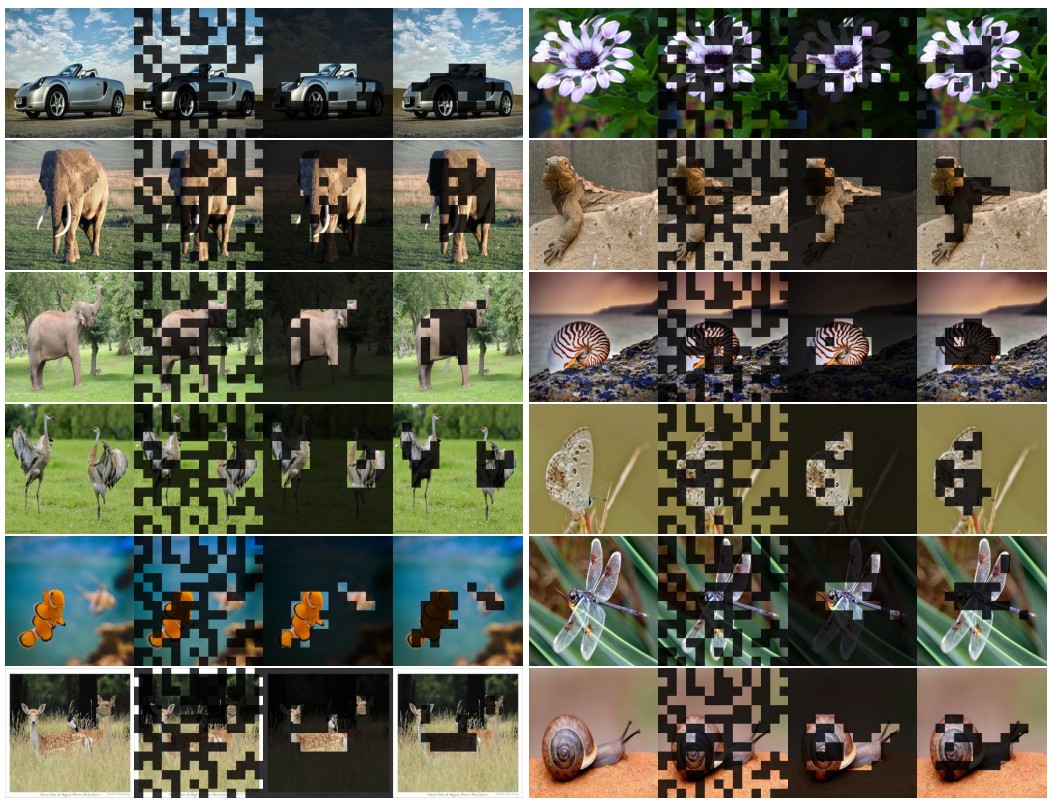

Figure 27: Visualizations for our three PatchDrop occlusion strategies: original, random (50% w.r.t the image), non-salient (50% w.r.t the forground predicted by DINO), and salient (50% of the backgrond as predicted by DINO) PatchDrop (shown from *left* to *right*). DeiT-B model achieves accuracies of 81.7%, 75.5%, 68.1%, and 71.3% across the ImageNet val. set for each level of occlusion illustrated from *left* to *right*, respectively.

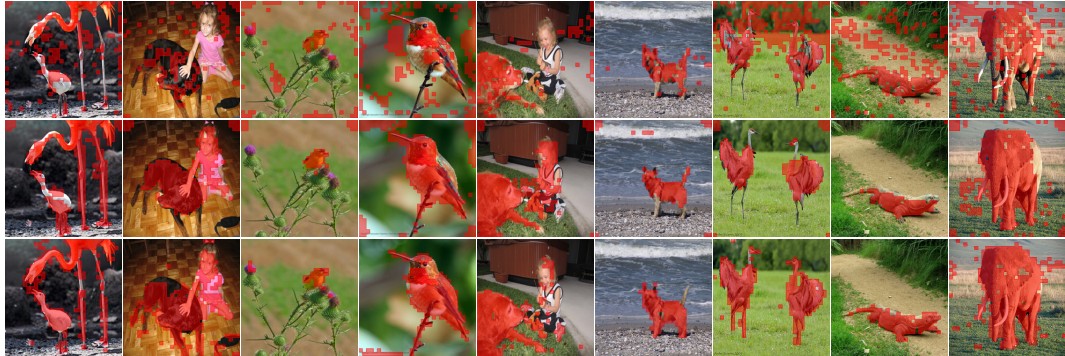

Figure 28: Automatic segmentation of images using class-token attention for a DeiT-S model. Original, SIN trained, and SIN distilled model outputs are illustrated from *top* to *bottom*, respectively.

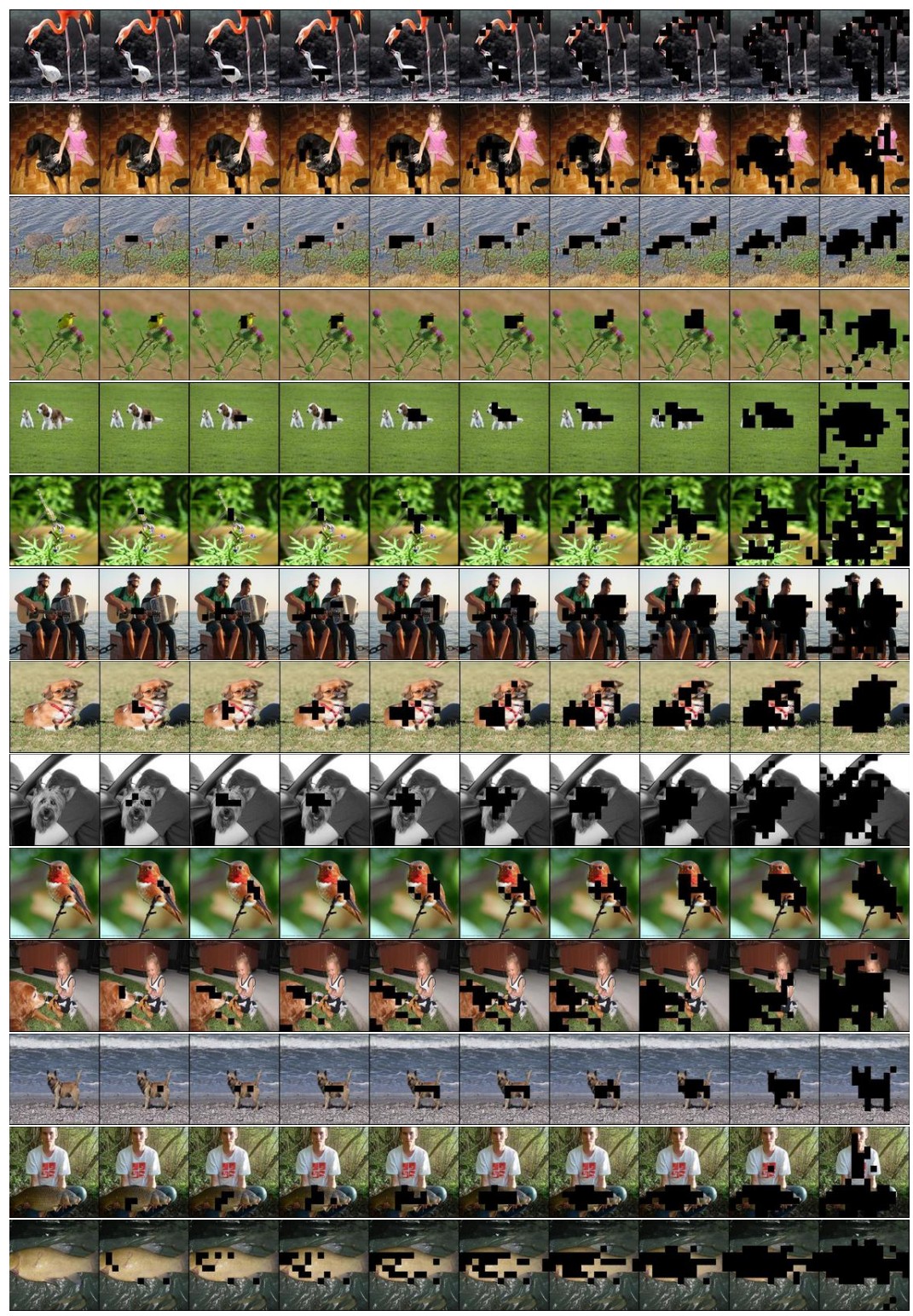

Figure 29: The variation (level increasing from *left* to *right*) of Salient PatchDrop on different images.

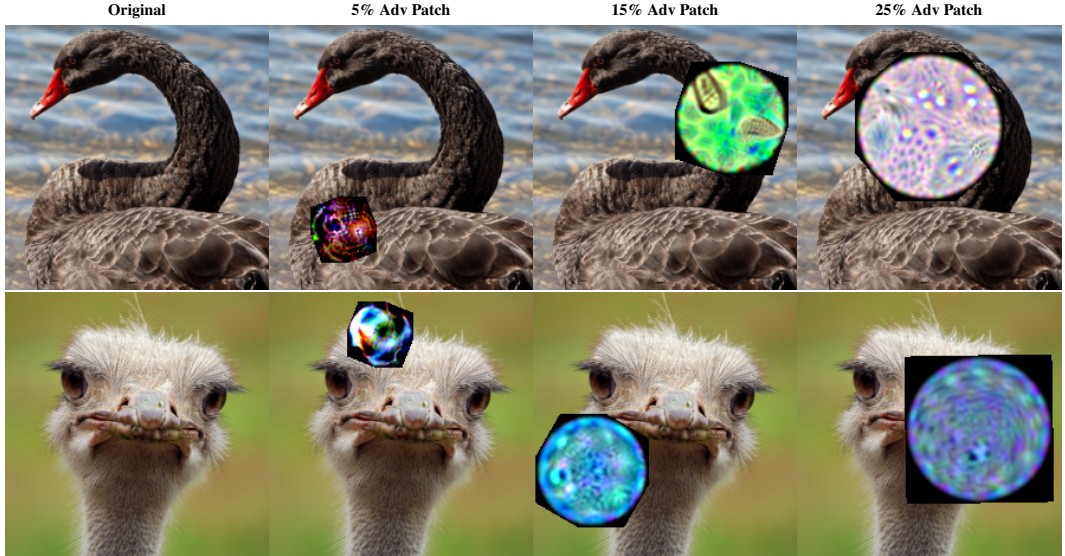

Figure 30: Adversarial patch (universal and untargeted) visualizations. *Top* row shows adversarial patches optimized to fool DeiT-S trained on ImageNet, while *bottom* row shows patches for DeiT-S-SIN. DeiT-S performs significantly better than DeiT-S-SIN. On the other hand, DeiT-SIN has higher shape-bias than DeiT-S.

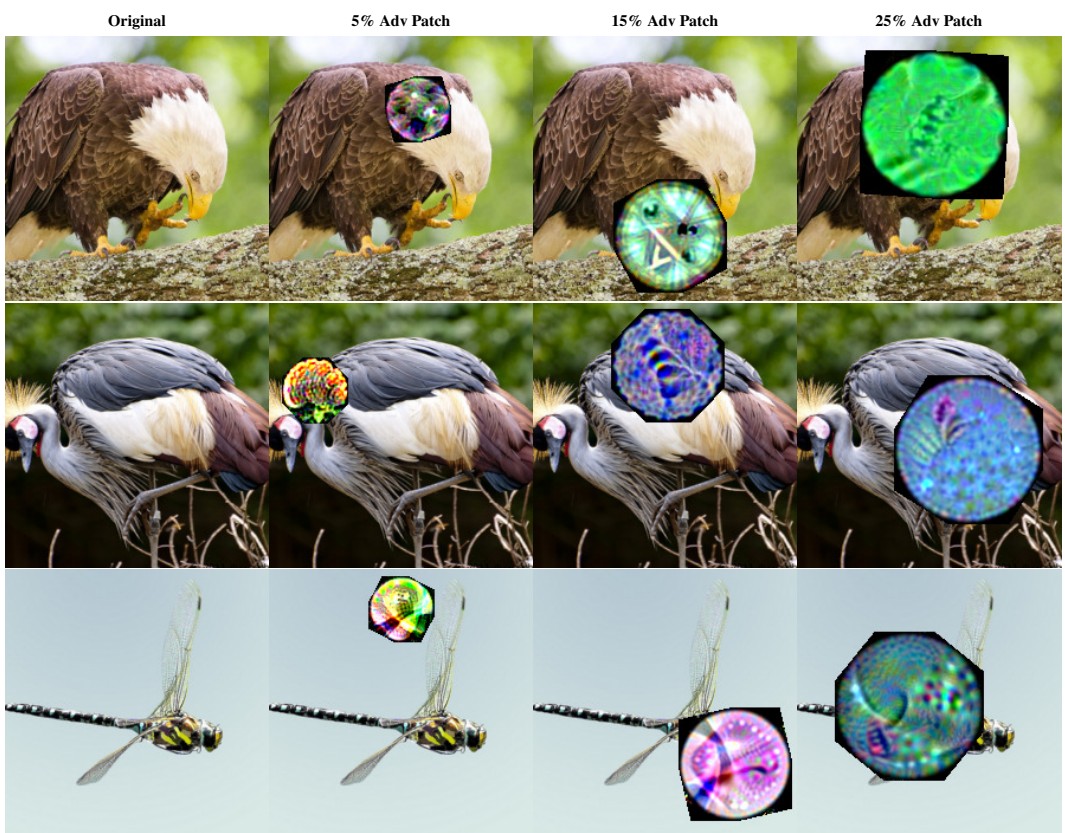

Figure 31: Adversarial patches (universal and untargeted) optimized to fool DeiT-T, DeiT-B, and T2T-24 models from *top* to *bottom*. These ViT models are more robust to such adversarial patterns than CNN (e.g., ResNet50).