# OpenReview forum: "Intriguing Properties of Vision Transformers"
_NeurIPS.cc/2021/Conference — NeurIPS 2021 Spotlight_

### Official Review · Reviewer_C6Em · 2021-07-12

**Rating:** 7
**Confidence:** 4

**Summary:**

This paper analyzes several intriguing properties of vision transformers (ViTs): First, it shows vision transformers have strong robustness against severe occlusions for foreground objects, non-salient background regions, and random patch locations, which are significantly better than the CNN counterparts. Second, vision transformers demonstrate better awareness of the shape whereas CNNs make decisions mainly based on texture. A shape distillation strategy is further proposed to improve the ViTs for a balance between classification performance and shape bias. Third, vision transformers achieve better robustness against spatial patch-level permutations, adversarial perturbations, and common natural corruptions. Finally, vision transformers transfer better than CNNs on visual classification and few-shot learning. A training scheme using multi-block class tokens (and/or average patch tokens) as a feature ensemble is applied to further boost the performance.

**Ethical Concerns:**

I think there is no significant ethical issue with this paper.

**Limitations And Societal Impact:**

Yes, the authors have mentioned the risk of the potential bias in the ImageNet pretrained ViTs and provided solutions.

**Main Review:**

Strengths / Contribution:

This paper provides a comprehensive analysis of the properties of vision transformers, mainly focusing on robustness, shape/texture awareness, and transferability.

1. In terms of the robustness evaluation, it is observed that ViTs (DeiT, ViT) are generally more robust than CNNs (ResNet, VGG, DenseNet, SqueezeNet) on three types of occlusions: Foreground objects, non-salient background regions, random patch locations. A more thorough and solid analysis in the supplementary shows that the better robustness is observed on various grid settings (from 4x4 to 16x16), which does not need to match the input grid (14x14) in the model (although the matched grid leads to a larger gap between ViTs and CNNs). This observation is interesting since it shows each token can still extract partially meaningful information from occluded patches and contribute to reasonable classification performance. Additionally, experiments on correlation coefficients also (partially) demonstrate that the class token in ViTs retains more information than the features in ResNet after applying occlusion to input.

     Moreover, some preliminary results on the adversarial patch attack also pinpoint that ViTs could have higher robustness than CNNs against untargeted, universal adversarial patches.

2. This paper examines the shape-bias of ViTs and CNNs and finds out the ViTs have a better awareness of shape in decisions, even with an explicit shape-oriented training scheme (i.e. use stylized ImageNet). To have a better balance between the classification performance and shape bias, the shape distillation using an auxiliary shape token is proposed. The shape distillation also helps the automated object segmentation task.

3. The authors attempt to validate if the position encoding in ViTs can preserve the global image context (e.g. global shape structure) by shuffling the image patches. However, I think this section is not well organized, as mentioned in the weaknesses section below.

4. Since ViTs naturally have multiple class tokens from all blocks, this paper proposes a feature ensemble approach that combines several class tokens for better generalization. Evaluations on transfer learning tasks including visual classification and few-shot learning demonstrate that using top several class tokens can greatly improve the feature transfer performance on a wide range of tasks.

In addition, the paper is well written, properly organized, and easy to read.

Weaknesses:
1. In Line 138, the paper uses correlation coefficient to measure the amount of preserved information in class token. However, the detailed calculation of this coefficient is not provided. In addition, it mentions "In the case of ResNet50, we consider features before the logit layer", where I am not sure if the features in ResNet are comparable with the class token (are both the feature vectors before the last linear layer for classification?)

2. In Sec. 3.3, the authors would like to "analyze if the sequence order modeled by positional encoding allows ViT to excel under occlusion handling". However, a simple comparison of ViT and ViT (w/o position encoding) on different occlusion settings is missing. In addition, the shuffling patches experiments suggest that transformers demonstrate high permutation invariance to the patch positions, and a claim "the effect of positional encoding towards injecting structural information of images models is limited" has been made. However, the Fig. 9 (right) shows a clear gap between ViT and ViT w/o pos, especially when the shuffle grid size (196) matches the input grid size, which seems contradictory to the claim.

3. In Sec. 3.4, it analyzes the robustness of ViTs under adversarial and natural perturbations and attempts to find its connection with shape-bias. However, only one certain type of adversarial attack is used (untargeted, universal adversarial patch in white-box setting). The conclusion would be more convincing if more adversarial attack methods are employed in the experiment.

4. Typo(s):

    Line 200: perofrm -> perform

    Line 276: specie classification -> species classification

    Figure 9: Y-axis is in %, thus the values should be 0, 10, 20, ..., 80 instead of 0, 0.1, ..., 0.8


**Time Spent Reviewing:**

4

---

> ### Author Response · Authors · 2021-08-10
> **Response**
>
> We thank the reviewer for the constructive comments.
>
>
> **1-a Correlation Coefficient:** We will add the exact formula for correlation coefficient in the paper as follows:
>
> $corr(x, y) = \frac{ \sum_i \hat{x_i} \hat{y_i} } {N},$
> where $\hat{x_i} = \frac {x_i - E[x_i]} {\sigma(x_i)}$, E[.] and $\sigma(\cdot)$ are mean and standard deviation operation [1]. In our case, random variables $x, y$ refer to the feature maps for an original and occluded image (defined over the entire ImageNet validation set).
>
> **1-b Clarity about Features:** Both feature vectors are taken before the last linear layer used for classification. ViT’s class token is the output from the last ViT block before the classifier just as ResNet50 features are before the last linear (classification) layer.
>
> **2-a ViT with and w/o Positional Encoding:** As suggested, we analyze Deit-T with and without positional encoding (Tables A-1, A-2 and A-3) against our proposed occlusions. Their behavior is similar to one another and follows the trend reported in Figure 3 of our manuscript.
>
> | Models        | IL:0% | IL:10% | IL:20% | IL:30% | IL:40% | IL:50% | IL:60% | IL:70% | IL:80% | IL:90% |
> | ------------- | ----- | ------ | ------ | ------ | ------ | ------ | ------ | ------ | ------ | ------ |
> | Deit-T        | 72.1  | 71.1   | 70.0   | 68.4   | 66.3   | 63.5   | 59.4   | 52.7   | 42.6   | 25.0   |
> | Deit-T-no-pos | 68.3  | 67.4   | 66.2   | 64.6   | 62.4   | 59.7   | 56.3   | 50.1   | 41.4   | 26.2   |
>
> Table A-1: Random PatchDrop. Top-1 (%) accuracy on ImageNet validation set against information loss (IL).
>
> | Models        | IL:0% | IL:10% | IL:20% | IL:30% | IL:40% | IL:50% | IL:60% | IL:70% | IL:80% | IL:90% |
> | ------------- | ----- | ------ | ------ | ------ | ------ | ------ | ------ | ------ | ------ | ------ |
> | Deit-T        | 72.1  | 67.9   | 61.7   | 54.7   | 47.3   | 38.9   | 30.1   | 21.1   | 12.1   | 4.3    |
> | Deit-T-no-pos | 68.3  | 63.2   | 56.7   | 50.1   | 43.1   | 35.7   | 28.3   | 20.5   | 12.9   | 5.9    |
>
> Table A-2: Salient PatchDrop. Top-1 (%) accuracy on ImageNet validation set against information loss (IL).
>
> | Models        | IL:0% | IL:10% | IL:20% | IL:30% | IL:40% | IL:50% | IL:60% | IL:70% | IL:80% | IL:90% |
> | ------------- | ----- | ------ | ------ | ------ | ------ | ------ | ------ | ------ | ------ | ------ |
> | Deit-T        | 72.1  | 70.9   | 69.4   | 67.6   | 64.8   | 61.1   | 56.2   | 49.6   | 38.6   | 22.1   |
> | Deit-T-no-pos | 68.3  | 67.0   | 65.6   | 63.6   | 61.1   | 58.0   | 53.6   | 47.2   | 37.3   | 22.6   |
>
> Table A-3: Non-salient PatchDrop. Top-1 (%) accuracy on ImageNet validation set against information loss (IL).
>
> **2-b Permutation Invariance Gap with and w/o Positional Encoding (Fig. 9):** When including positional encoding in ViTs, one might expect to remain sensitive to the sequence ordering (i.e., the order of image patches). However, we note that the performance of current ViTs with positional encodings remains much higher even w.r.t CNNs (like ResNet50) which experience a significant drop in performance under permutations.
>
> We agree that there is a gap in performance between ViT with and without positional encoding (Figure 9, right plot). This indicates that having positional encoding makes ViT more sensitive to change in global image structure, such as patch shuffle (a desirable property), in comparison to no position encoding. However, the effect of positional encoding towards preserving global structure is relatively weak, compared to CNNs. As we increase the number of patches and shuffle, the performance is expected to drop significantly as meaningful global image structure is being destroyed. We will improve the discussion to clarify our analysis in the revised manuscript.
>
> **3 Other Adversarial Attacks:** As recommended, we evaluated the adversarial robustness of these models via FGSM [2], PGD [3] and MIFGSM [4] attacks (white-box) at perturbation budget of eps 2,4,8 (Tables B-1, B-2 and B-3). ViTs perform favorably against ResNet. ViTs trained with augmentations show more robustness than ViTs trained on SIN without augmentations. Attacks are evaluated in white-box settings. We will add these results in the revised manuscript.
>
> | Models       | $\epsilon=2$ | $\epsilon=4$ | $\epsilon=8$ |
> | ------------ | ------------ | ------------ | ------------ |
> | ResNet50     | 7.6          | 6.6          | 6.1          |
> | Deit-T       | 18.2         | 13.8         | 11.0         |
> | Deit-S       | 42.2         | 36.6         | 31.6         |
> | Deit-B       | 51.8         | 46.6         | 41.9         |
> | ResNet50-SIN | 4.0          | 3.1          | 3.1          |
> | Deit-S-SIN   | 3.9          | 2.4          | 1.6          |
> | Deit-T-SIN   | 0.6          | 0.3          | 0.2          |
>
> Table B-1: Robustness against FGSM ($l_\infty$). Top-1 (%) accuracy on ImageNet validation set. $\epsilon$ represents the perturbation budget by which each pixel is changed in the image.
>
> | Models       | $\epsilon=2$ | $\epsilon=4$ | $\epsilon=8$ |
> | ------------ | ------------ | ------------ | ------------ |
> | ResNet50     | 0.4          | 0.1          | 0.0          |
> | Deit-T       | 4.0          | 0.8          | 0.0          |
> | Deit-S       | 12.6         | 3.3          | 0.5          |
> | Deit-B       | 15.7         | 5.3          | 1.3          |
> | ResNet50-SIN | 0.3          | 0.1          | 0.0          |
> | Deit-S-SIN   | 0.2          | 0.0          | 0.0          |
> | Deit-T-SIN   | 0.0          | 0.0          | 0.0          |
>
> Table B-2: Robustness against MIFGSM  with 5 iterations ($l_\infty$). Top-1 (%) accuracy on ImageNet validation set. $\epsilon$ represents the perturbation budget by which each pixel is changed in the image.
>
> | Models       | $\epsilon=2$ | $\epsilon=4$ | $\epsilon=8$ |
> | ------------ | ------------ | ------------ | ------------ |
> | ResNet50     | 0.2          | 0.0          | 0.0          |
> | Deit-T       | 1.5          | 0.1          | 0.0          |
> | Deit-S       | 8.0          | 1.0          | 0.2          |
> | Deit-B       | 15.5         | 6.9          | 2.2          |
> | ResNet50-SIN | 0.2          | 0.0          | 0.0          |
> | Deit-S-SIN   | 0.5          | 0.0          | 0.0          |
> | Deit-T-SIN   | 0.0          | 0.0          | 0.0          |
>
> Table B-3: Robustness against PGD  with 5 iterations ($l_\infty$). Top-1 (%) accuracy on ImageNet validation set. $\epsilon$ represents the perturbation budget by which each pixel is changed in the image.
>
> As recommended, we will fix all typos in the revised manuscript.
>
> **References**\
> [1] David Forysth, “Probability and Statistics for Computer Science”, P39, Sec 2.21 \
> [2] Goodfellow et.al “Explaining and Harnessing Adversarial Examples.”, ICLR, 2014 \
> [3] Madry et.al “Towards Deep Learning Models Resistant to Adversarial Attacks”, ICLR, 2018 \
> [4] Dong et.al “Boosting Adversarial Attacks with Momentum”, CVPR, 2018

---

> > ### Comment · Reviewer_C6Em · 2021-08-29
> > **Final Comments**
> >
> > I think my concerns are well addressed by the authors' response. Thus, I would like to keep my rating and wish the authors could revise the paper as mentioned in the response.

---

### Official Review · Reviewer_yAfc · 2021-07-15

**Rating:** 7
**Confidence:** 4

**Summary:**

This paper explores a variety of properties of ViTs about occlusions, shapes and texture bias, perturbations, robustness and domain shifts. The authors empirically demonstrate above merits of ViTs compared with CNNs by sufficient experiments on fifteen vision datasets. Besides, an architectural modification to DeiT for shape distillation and a transfer approach utilizing an ensemble of representations with a pre-trained ViTs are proposed as new design choices.

**Limitations And Societal Impact:**

The authors do provide description of the limitations mentioned and potential negative societal impact of their work.

**Main Review:**

### Originality：
This paper shows some intriguing properties of ViTs. Besides, the authors also propose two design choices for ViTs for different tasks. Although some properties are intriguing, the experiment design in this paper basically follows the discussion of these properties of CNNs, which leads to the lack of novelty.

### Quality
The submission is technically sound and is a complete piece of work. It also provides attention maps for explanation. However, only experimental explanations for these properties are given, and theoretical proof is lacking.

### Clarity
The paper is well organized with clear tables and figures and easy to follow.

### Significance:
This paper shows some intriguing properties of ViTs, which may be conducive to further understanding of this new architecture. The amount of experiments is sufficient, the arrangement of experiments is reasonable, and the results are convincing.

As mentioned in line 234, you attribute the robustness of ViT to its flexible and dynamic receptive field by exclusion. I would like to know if you can provide direct experiments to prove this conclusion.


**Time Spent Reviewing:**

12

---

> ### Author Response · Authors · 2021-08-10
> **Response**
>
> We thank the reviewer for the constructive comments.
>
> **Novelty:** In this work, we conduct a thorough analysis of ViTs by covering three transformer families (ViT, DeiT, T2T) across 15 vision datasets. As commented by Anonymous Reviewer Xmdv: “While existing papers regarding transformers in vision focus on the architectural novelty or applying them to novel tasks, this paper provides a thorough and extensive study on the properties of vision transformers regarding robustness.” Our analysis has led to several interesting insights, which we believe will be significant and helpful to the community.
>
> In addition to our extensive empirical analysis and new findings, we introduce several novel design choices to highlight the strong potential of ViTs. To this end, we propose an architectural modification to DeiT to encode shape-information via a dedicated token that demonstrates how seemingly contradictory cues can be modeled with distinct tokens within the same architecture (Section 3.2), leading to favorable implications such as automated segmentation without pixel-level supervision. Furthermore, our off-the-shelf feature transfer approach utilizes an ensemble of representations derived from a single architecture to obtain state-of-the-art generalization with a pre-trained ViT (Figure 1 and Section 3.5).
>
> **Dynamic Receptive field:** We further study the ViT behavior to focus on the informative signal regardless of its position. In our new experiment, during inference, we rescale the input image to 128x128 and place it within black background of size 224x224. In other words, rather than removing or shuffling image patches, we reflect all the image information into few patches. We then move the position of these patches to the upper/lower right and left corners of the background. On average, Deit-S shows 62.9% top-1 classification accuracy and low variance  (62.9 $\pm$ 0.05). In contrast, ResNet50 achieves only 5.4% top-1 average accuracy. These results suggest that ViTs can exploit discriminative information regardless of its position (Table A). We will add these results and a discussion in the revised manuscript. We will also add a visualization depicting the change in attention, as the image is moved within the background. Further, a similar behavior for the dynamic receptive field of ViTs is also presented in Figure 4 of our manuscript and was inspired by [1] which compares the receptive field of self-attention with that of CNNs.
>
> | Models    | top-right | top-left | bottom-right | bottom-left |
> | --------- | --------- | -------- | ------------ | ----------- |
> | DeiT-T    | 51.21     | 51.38    | 50.61        | 50.70       |
> | DeiT-S    | 63.14     | 63.01    | 62.62        | 62.79       |
> | DeiT-B    | 69.37     | 69.29    | 69.18        | 69.20       |
> | ResNet-50 | 5.59      | 5.71     | 4.86         | 5.30        |
>
> Table A: Top-1 (%) accuracy on ImageNet validation set.
>
> As recommended, We will fix the units in Figures 9 and 10 and make all plots consistent throughout the paper.
>
> **References**\
> [1] Cordonnier et.al “On the Relationship between Self-Attention and Convolutional Layers.”, ICLR, 2020

---

> > ### Comment · Reviewer_yAfc · 2021-09-10
> > **Final comments**
> >
> > I'd like to keep my rating and recommend the submission for acceptance.

---

### Official Review · Reviewer_KJfC · 2021-07-16

**Rating:** 7
**Confidence:** 4

**Summary:**

The paper systematically studies ViT’s performance on occlusions, domain shifts, spatial permutations, adversarial and natural perturbations. Experiments show that ViT models are relatively more robust to occlusions, perturbations and domain shifts, and less sensitive to local textures. Off-the-shelf features from ViT models yield high accuracy on several downstream classification datasets.

**Limitations And Societal Impact:**

Yes

**Main Review:**

Overall, the paper is clear and well-written. Experiments are thorough and fair. I don’t see any red flags during my review, except for a few questions and suggestions as follows:
-	L190 the authors discovered that shape biased ViT offers automated object segmentation. A recent work, [1], shows this property as well by self-supervised training without any labels. Exploring or discussing this more would be good.
-	L138, the authors claimed that “class token preserves information”. It is a confusing statement as the authors may be simply stating a fact, or it could be that the authors argued the class token itself was essential. Either way it is somewhat inaccurate. The experimented architecture is (a) ViT + class token, and (b) CNN + feature map. If (a) is better than (b), it’s obvious that it doesn’t imply class token is better than feature map.
-	All CNN experiments are done with ResNet50. However, ViT models are the latest cool models whereas ResNet was proposed 6 years ago. Using state-of-the-art CNNs, some of which are highly similar to ResNet but yield much better accuracy, such as RegNetY [2], is strongly encouraged.
-	Table 7 studies transfer accuracy using off-the-shelf features from various ViT stages. Intuitively, of course, the final layer’s features are the best ones, but the study is only done on CUB dataset, which is notorious for being not reliable. I would suggest the authors to make the study more rigorous, however small or intuitive the study’s result may yield.
-	In the y axis of figures, some of them use percentage (e.g., 50% accuracy), while some of them use decimal (e.g., 0.5 accuracy). Please make it consistent over the whole paper. Figure 9 and 10 have the wrong unit, i.e., 0.5% accuracy.


[1] Emerging Properties in Self-Supervised Vision Transformers, Caron et al.
[2] Designing Network Design Spaces, Radosavovic et al.


**Time Spent Reviewing:**

1.5

---

> ### Author Response · Authors · 2021-08-10
> **Response**
>
> We thank the reviewer for the positive feedback.
>
> **Discussion on DINO (L190):** We discussed DINO [1] in Section 2 and Table 4 of our manuscript. DINO encodes shape information by matching local image views with its global structure to perform auto-segmentation. On the other hand, our approach performs auto-segmentation by learning from stylized augmentations (textureless information) either via distillation or direct supervision (Table 4 in our manuscript). Both DINO and our work highlight that ViTs trained in a supervised setting on natural images struggle to provide favorable auto-segmentation results. In our work, we further found that ViTs auto-segmentation property stems from their ability to encode shape information. We believe that integrating both our approach and DINO is worth exploring in the future.  To highlight few open research questions:  a) Can self-supervision on stylized ImageNet (SIN) improve segmentation ability of DINO?, and b) Can modifying the DINO training such as texture (IN) based local views and shape (SIN) based global views enhances DINO ability to auto-segment? These research questions are beyond the scope of the current study, however, we will add a discussion on these aspects in the revised manuscript as possible future work. As recommended by the reviewer, we will add a detailed discussion with DINO in the revised manuscript.
>
> **Class token or Feature (L138):**   The discussion on L138 is about the robustness comparison of ViT features with CNNs in the presence of information loss. In our experimental settings, we used ViTs with class tokens that interact with patch tokens throughout the network and are subsequently used for classification. We agree that not all ViT designs use a class token e.g., Swin Transformer [3] uses an average of all tokens. To this end, we conduct new experiments (Tables A-1, A-2 and A-3) using three variants of the recent Swin Transformer [3]. Our results show the robustness of ViT architecture against information loss. We will replace “Class Token Preserves Information” to “ViT representations are Robust against Information Loss” at L138 to improve clarity in the revised manuscript.
>
> | Models   | IL:0% | IL:10% | IL:20% | IL:30% | IL:40% | IL:50% | IL:60% | IL:70% | IL:80% | IL:90% |
> | -------- | ----- | ------ | ------ | ------ | ------ | ------ | ------ | ------ | ------ | ------ |
> | ResNet50 | 76.0  | 45.2   | 17.7   | 6.4    | 2.1    | 0.84   | 0.43   | 0.27   | 0.23   | 2.1    |
> | Swin-T   | 80.9  | 80.2   | 79.2   | 77.9   | 76.1   | 73.4   | 70.2   | 64.6   | 55.5   | 37.5   |
> | Swin-S   | 82.9  | 82.2   | 81.3   | 80.0   | 77.9   | 75.2   | 71.1   | 64.1   | 54.7   | 38.6   |
> | Swin-B   | 84.8  | 84.4   | 83.3   | 82.0   | 80.3   | 78.1   | 75.1   | 70.2   | 60.7   | 43.2   |
>
> Table A-1: Random PatchDrop. Top-1 (%) accuracy on ImageNet validation set against information loss (IL).
>
> | Models   | IL:0% | IL:10% | IL:20% | IL:30% | IL:40% | IL:50% | IL:60% | IL:70% | IL:80% | IL:90% |
> | -------- | ----- | ------ | ------ | ------ | ------ | ------ | ------ | ------ | ------ | ------ |
> | ResNet50 | 76.0  | 47.9   | 25.5   | 12.9   | 6.7    | 3.5    | 1.9    | 1.1    | 0.6    | 0.3    |
> | Swin-T   | 80.9  | 78.9   | 76.2   | 72.3   | 67.1   | 60.2   | 51.6   | 40.7   | 26.6   | 9.1    |
> | Swin-S   | 82.9  | 80.9   | 78.0   | 74.1   | 68.8   | 62.1   | 54.2   | 43.7   | 30.8   | 13.6   |
> | Swin-B   | 84.8  | 83.0   | 80.6   | 77.0   | 72.7   | 66.6   | 58.6   | 48.6   | 35.7   | 16.8   |
>
> Table A-2: Salient PatchDrop. Top-1 (%) accuracy on ImageNet validation set against information loss (IL).
>
> | Models   | IL:0% | IL:10% | IL:20% | IL:30% | IL:40% | IL:50% | IL:60% | IL:70% | IL:80% | IL:90% |
> | -------- | ----- | ------ | ------ | ------ | ------ | ------ | ------ | ------ | ------ | ------ |
> | ResNet50 | 76.0  | 68.6   | 59.0   | 47.4   | 35.4   | 24.0   | 14.8   | 7.7    | 3.1    | 0.82   |
> | Swin-T   | 80.9  | 80.2   | 79.2   | 77.8   | 76.1   | 73.4   | 69.8   | 64.3   | 55.4   | 37.8   |
> | Swin-S   | 82.9  | 82.3   | 81.5   | 80.2   | 78.6   | 76.2   | 72.6   | 67.2   | 57.4   | 39.1   |
> | Swin-B   | 84.8  | 84.3   | 83.8   | 83.0   | 81.5   | 79.5   | 76.6   | 72.0   | 63.7   | 46.6   |
>
> Table A-3: Non-salient PatchDrop. Top-1 (%) accuracy on ImageNet validation set against information loss (IL).
>
> **On RegNetsY [2]:** As recommended by the reviewer, we evaluated three variants of RegNetY against our proposed occlusions (Tables B-1, B-2 and B-3). RegNetY shows relatively higher robustness when compared to ResNet50, but overall behaves similar to other CNN models. We will add these new results with RegNetY in the revised manuscript.
>
> | Models     | IL:0% | IL:10% | IL:20% | IL:30% | IL:40% | IL:50% | IL:60% | IL:70% | IL:80% | IL:90% |
> | ---------- | ----- | ------ | ------ | ------ | ------ | ------ | ------ | ------ | ------ | ------ |
> | RegNetY-4  | 79.2  | 59.9   | 37.9   | 22.2   | 10.3   | 4.4    | 1.7    | 0.7    | 0.4    | 0.4    |
> | RegNetY-8  | 79.8  | 67.0   | 49.9   | 31.2   | 15.7   | 7.4    | 3.3    | 1.6    | 0.9    | 0.6    |
> | RegNetY-16 | 80.2  | 65.8   | 47.5   | 30.2   | 15.6   | 6.9    | 3.1    | 1.3    | 0.6    | 0.3    |
> | Deit-S     | 79.8  | 79.1   | 77.6   | 76.3   | 73.9   | 70.9   | 66.9   | 60.3   | 50.4   | 31.1   |
>
> Table B-1: Random PatchDrop. Top-1 (%) accuracy on ImageNet validation set against information loss (IL).
>
> | Models     | IL:0% | IL:10% | IL:20% | IL:30% | IL:40% | IL:50% | IL:60% | IL:70% | IL:80% | IL:90% |
> | ---------- | ----- | ------ | ------ | ------ | ------ | ------ | ------ | ------ | ------ | ------ |
> | RegNetY-4  | 79.2  | 60.3   | 40.3   | 23.8   | 13.3   | 7.1    | 3.7    | 2.0    | 1.2    | 0.6    |
> | RegNetY-8  | 79.8  | 63.8   | 44.8   | 28.4   | 16.7   | 9.5    | 5.2    | 2.7    | 1.4    | 0.6    |
> | RegNetY-16 | 80.2  | 64.1   | 45.7   | 28.7   | 16.4   | 8.8    | 4.4    | 2.0    | 0.9    | 0.4    |
> | Deit-S     | 79.8  | 76.7   | 71.9   | 65.9   | 59.0   | 50.4   | 40.9   | 29.5   | 17.4   | 5.7    |
>
> Table B-2: Salient PatchDrop. Top-1 (%) accuracy on ImageNet validation set against information loss (IL).
>
> | Models     | IL:0% | IL:10% | IL:20% | IL:30% | IL:40% | IL:50% | IL:60% | IL:70% | IL:80% | IL:90% |
> | ---------- | ----- | ------ | ------ | ------ | ------ | ------ | ------ | ------ | ------ | ------ |
> | RegNetY-4  | 79.2  | 73.9   | 67.6   | 59.7   | 50.3   | 39.8   | 28.9   | 18.3   | 9.1    | 2.6    |
> | RegNetY-8  | 79.8  | 76.1   | 70.8   | 63.7   | 54.7   | 44.2   | 32.7   | 21.6   | 11.4   | 3.7    |
> | RegNetY-16 | 80.2  | 75.8   | 70.4   | 63.5   | 55.2   | 45.3   | 34.2   | 22.8   | 12.2   | 3.8    |
> | Deit-S     | 79.8  | 78.9   | 77.8   | 76.0   | 73.9   | 70.8   | 66.3   | 59.4   | 48.3   | 29.9   |
>
> Table B-3: Non-salient PatchDrop. Top-1 (%) accuracy on ImageNet validation set against information loss (IL).
>
> **More Ablations:** As recommended, we extended ablations presented in Table 7 of our manuscript to other datasets: Flowers (small dataset) and the large-scale iNaturalist.  Table C shows the results on both these datasets. We observe that the transferability trends remain similar as most discriminative tokens from the last four blocks of Deit-S achieve better transfer accuracy, compared to only using the last token. We find some exception to this on the Flower dataset where using tokens from all blocks have relatively better improvement (only 1.2%), compared to tokens from the last four blocks. It is worth mentioning that concatenating tokens from all blocks also increases the number of parameters e.g., transfer to Flowers from all tokens has 3x more learnable parameters than using only the last four tokens. We will update Table 7 in the revised manuscript.
>
> | Blocks                 | Class Tokens | Patch Tokens | Flowers   | iNaturalist |
> | ---------------------- | ------------ | ------------ | --------- | ----------- |
> | Only 12th (last block) | &#10004;      | x            | 82.58     | 38.28       |
> | Only 12th (last block) | &#10004;      | &#10004;      | 86.58     | 41.22       |
> | From 1th to 12th       | &#10004;     | x            | 90.00     | **45.15**   |
> | From 1th to 12th       | &#10004;     | &#10004;      | 90.33     | 45.12       |
> | From 9th to 12th       | &#10004;      | x            | **91.38** | 44.03       |
> | From 9th to 12th       | &#10004;      | &#10004;      | 91.27     | 43.33       |
>
> Table C: Ablative Study for off-the-shelf feature transfer on Flowers and iNaturalist datasets using ImageNet pre-trained DeiT-S.
>
> As recommended, we will fix the units in Figures 9 and 10 and make all plots consistent.
>
> **References** \
> [1] Caron et. al “Emerging Properties in Self-Supervised Vision Transformers.” ArXiv, 2021 \
> [2] Radosavovic et.al “Designing Network Design Spaces.” CVPR, 2020 \
> [3] Liu et.al “Swin Transformer: Hierarchical Vision Transformer using Shifted Windows.”, ArXiv, 2021

---

> > ### Comment · Reviewer_KJfC · 2021-08-17
> > **Final Evaluation**
> >
> > I'd like to thank the authors for address all of my questions. I'm overall satisfied with the feedback and am confident that the submission is a good work and thus should warrant acceptance. I would like to keep my initial rating and recommend the submission for acceptance.

---

### Official Review · Reviewer_Xmdv · 2021-07-23

**Rating:** 7
**Confidence:** 4

**Summary:**

This paper analyzes intriguing properties of Vision Transformers (ViTs) in the context of robustness and generalizability compared to CNNs. In particular, they show that (i) ViTs are highly robust to severe occlusions and perturbations, (ii) ViTs are significantly less biased towards texture, (iii) ViTs that encode shape representation provides accurate semantic segmentation without labels, (iv) we can ensemble off-the-shelf tokens from a single ViT model for transfer learning to a wide range of tasks. They conduct their experiments on three transformer families (ViT, DeiT, T2T) across 15 vision datasets.


**Limitations And Societal Impact:**

I appreciate that the authors pointed out that their analysis is based on “ImageNet (ILSVRC ‘12)”  pre-trained vision transformer which suffer its own biases (e.g. dominantly Western) as well as privacy risks. Therefore, their in-depth analysis is limited to the scope of this training set and may very well reflect the potential biases in the learned representations.

**Main Review:**

While existing papers regarding transformers in vision focus on the architectural novelty or applying them to novel tasks, this paper provides a thorough and extensive study on the properties of vision transformers regarding robustness.

I find it particularly intriguing that a single ViT can model conflicting features (e.g. both texture and shape bias) using separate tokens, which is a property that CNNs cannot achieve. This property can be particularly useful in increasing the downstream transferability to a more wide range of tasks. Not all downstream tasks require the same set of invariances, e.g. invariance to rotation can be beneficial in view-independent aerial image recognition but harmful in tasks such as detecting which way is up in a photograph [1]. In these cases, we could simply ensemble different blocks whose tokens encode different invariances, as how others did in Section 3.5.

The paper is well-written and clear except for minor typos: (271) comapre -> compare.

Overall, I believe this is a solid work that will be helpful to the community.



**Time Spent Reviewing:**

14

---

> ### Author Response · Authors · 2021-08-10
> **Response**
>
> We thank the reviewer for the thorough and extensive review of our paper as well as all the encouraging and positive remarks. We will fix the typos in the revised manuscript.
>
> **Limitations And Societal Impact:** \
> As suggested, we will discuss the societal impact on using the ImageNet for the analysis presented in this work. We will highlight the potential issues regarding data bias, although attempts to address this issue is outside the scope of this work.

---

> > ### Comment · Reviewer_Xmdv · 2021-09-02
> > **Final Comment**
> >
> > I would like to keep my rating and recommend the submission for acceptance.

---

### Decision · Program_Chairs · 2021-09-27

**Decision:**

Accept (Spotlight)

**Comment:**

The reviewers agree that this is a solid paper that should be accepted. The analysis of vision transformers vs convnets provided in the paper is through, interesting and potentially useful for practitioners. The authors mostly addressed the (not very numerous) reviewer' concerns. Clear accept.